# Systems approach identifies monocyte imbalance in symptomatic and asymptomatic *P. vivax* malaria

Stephanie I Studniberg [1], Mariam Bafit[1], Lisa J Ioannidis[2,3], Matthew J Worley[1], Leily Trianty[4,5], Retno A S Utami[5], Agatha M Puspitasari[5], Dwi Apriyanti[4], Farah N Coutrier[4,5], Jeanne R Poespoprodjo [6,7,8,9], Enny Kenangalem[6,7], Benediktus Andries [6], Pak Prayoga[6], Ric N Price [9,10,11], Rintis Noviyanti[4,5], Alexandra L Garnham[2,3] & Diana S Hansen [1]✉

## Abstract

**Although asymptomatic malaria was historically perceived as innocuous, emerging evidence revealed an immunosuppressive signature induced by asymptomatic *Plasmodium falciparum* infections. To examine if a similar process occurs in *Plasmodium vivax* malaria, we pursued a systems approach, integrating transcriptional profiling together with previously reported and novel mass cytometry phenotypes from individuals with symptomatic and asymptomatic *P. vivax* malaria. Symptomatic *P. vivax* malaria featured upregulation of anti-inflammatory pathways and checkpoint receptors. A profound downregulation of transcripts with roles in monocyte function was observed in symptomatic *P. vivax* malaria. This reduction in monocyte transcriptional activity was accompanied by a significant depletion of CCR2+CXCR4+ classical monocytes in symptomatic individuals. Despite allowing transcriptional profiles supporting T-cell differentiation, dysregulation of genes associated with monocyte activation and the inflammasome was also evident in individuals carrying *P. vivax* asymptomatic infections. Our results identify monocyte dysregulation as a key feature of the response to *P. vivax* malaria and support the concept that asymptomatic infection is not innocuous and might not support all immune processes required to eliminate parasitemia or efficiently respond to vaccination.**

**Keywords** Asymptomatic Malaria; Monocytes; *Plasmodium vivax*; Transcriptional Profiling; Systems Immunology
**Subject Categories** Immunology; Microbiology, Virology & Host Pathogen Interaction

## Introduction

Despite significant reductions in case incidence and mortality, almost half the world's population is still at risk of malaria infection (World Health Organization, 2022). Whilst *Plasmodium falciparum* causes 95% of malaria cases in sub-Saharan Africa (World Health Organization, 2022), elsewhere, *Plasmodium vivax* is the most widely distributed malaria-causing parasite and is increasingly recognized as a significant barrier to elimination (Habtamu et al, 2022).

The development of immunity to malaria requires multiple *Plasmodium* re-infections, usually over the course of several years, with children in malaria-endemic areas suffering the major burden of severe morbidity and mortality (Marsh and Kinyanjui, 2006). Once achieved, immunity to malaria can reduce parasite density to below the threshold inducing clinical symptoms, and is therefore also referred to as "clinical immunity" (Doolan et al, 2009). Various epidemiological studies have reported that clinical immunity to malaria develops faster in response to *P. vivax* compared with *P. falciparum* parasites (Lin et al, 2010; Michon et al, 2007; Mueller et al, 2009). Moreover, severe clinical symptoms have been reported to peak earlier and decrease at a faster rate in communities experiencing high *P. vivax*, compared with *P. falciparum* transmission (Muller et al, 2009).

Seminal challenge studies involving the passive transfer of sera from adults with clinical immunity to *P. falciparum* malaria to unexposed participants have established immunoglobulin G (IgG) as an important component in reducing parasite burden and diminishing malaria symptoms (Cohen et al, 1961). Since studies have found antibodies to *P. vivax* blood-stage antigens to be positively associated with protection from symptomatic episodes (Franca et al, 2017; He et al, 2019; Stanisic et al, 2013). Circulating T helper ($T_H$)2-polarized T follicular helper ($T_{FH}$) cells—critical for providing B-cell help for antibody formation—were also reported to increase in frequency in response to *P. vivax* infection, and

[1]Monash Biomedicine Discovery Institute, Department of Microbiology, Monash University, Clayton, VIC, Australia. [2]Walter and Eliza Hall Institute of Medical Research, Parkville, VIC, Australia. [3]Department of Medical Biology, The University of Melbourne, Parkville, VIC, Australia. [4]Eijkman Research Centre for Molecular Biology, East Java, Indonesia. [5]Exeins Health Initiative, Jakarta, Indonesia. [6]Timika Malaria Research Program, Papuan Health and Community Development Foundation, Timika, Papua, Indonesia. [7]Mimika District Hospital and District Health Authority, Timika, Papua, Indonesia. [8]Centre for Child Health and Department of Child Health, Faculty of Medicine, Public Health and Nursing, Universitas Gadjah Mada, Yogyakarta, Indonesia. [9]Global and Tropical Health Division, Menzies School of Health Research, Charles Darwin University, Darwin, NT, Australia. [10]Mahidol-Oxford Tropical Medicine Research Unit, Faculty of Tropical Medicine, Mahidol University, Bangkok, Thailand. [11]Centre for Tropical Medicine, Nuffield Department of Medicine, University of Oxford, Oxford, UK. ✉E-mail: diana.hansen@monash.edu

continue to increase with subsequent exposures (Figueiredo et al, 2017; Ioannidis et al, 2021). Despite these protective features, clinical immunity to malaria is non-sterilizing, with adults living in malaria-endemic areas often experiencing continuous asymptomatic malaria infections (Langhorne et al, 2008).

Historically, asymptomatic malaria has been perceived as benign and even beneficial to maintain clinical immunity in populations living in malaria-endemic areas (Al-Yaman et al, 1997; Bereczky et al, 2007; Farnert et al, 1999; Sonden et al, 2015). However, mounting evidence has begun to unveil diverse detrimental impacts of asymptomatic malaria, including anemia (Pava et al, 2016), splenomegaly as a result of persistent filtration and retention of infected erythrocytes (Kho et al, 2021; Kho et al, 2024), placental malaria causing maternal anemia (Alemayehu et al, 2024; Gemechu et al, 2023), and increased susceptibility to secondary infections (Nyirenda et al, 2018). Aligned with those views, we have recently identified a strong blood transcriptional signature driving several immunosuppressive processes in a cohort of individuals carrying low parasite density asymptomatic *P. falciparum* malaria (Studniberg et al, 2022), suggesting that these subclinical infections might not support immune processes capable of fully controlling parasitemia or efficiently responding to malaria vaccines. Consistent with this hypothesis, studies conducted within the context of the PfSPZ-CVac trial that combines challenge with whole infectious *P. falciparum* sporozoites with accompanying antimalarial chemoprophylaxis found vaccine efficacy to improve dramatically when administered in the absence of blood-stage parasitemia (Diawara et al, 2024; Murphy et al, 2021).

Whether asymptomatic *P. vivax* malaria influences host transcriptional processes to induce immunosuppression or another deleterious consequence of infection remains to be elucidated. To address this knowledge gap, we pursued a systems biology approach integrating blood transcriptional profiling, clinical parameters, previously reported (Ioannidis et al, 2021) and new single-cell mass cytometry phenotypes of individuals experiencing symptomatic and asymptomatic *P. vivax* malaria. Our main findings revealed a strong downregulation of pro-inflammatory pathways, upregulation of checkpoint receptors, and profound inhibition of monocyte-associated transcripts during symptomatic *P. vivax* malaria. This reduction in monocyte transcriptional activity was accompanied by a significant depletion of CD10$^+$CCR2$^+$CXCR4$^+$ classical monocytes. Whilst transcriptional profiles supporting T-cell differentiation and effector function were enriched in asymptomatic *P. vivax* malaria, monocyte function was also downregulated in these persistent infections of low parasite burden. These data are consistent with recent findings in non-febrile *P. falciparum* malaria (Studniberg et al, 2022) and support the notion that asymptomatic *P. vivax* malaria is not innocuous and might not support all immune processes required to fully control parasitemia or efficiently respond to vaccination.

# Results

## Cohort characteristics

A cross-sectional study was conducted in the Timika region of Papua, Indonesia, as previously described (Ioannidis et al, 2021). Briefly, Papuans living in Timika reside in the lowlands, where malaria transmission occurs, and the highlands, where transmission is absent. Migration of adults without pre-existing immunity

to malaria from the highlands to the lowlands means any age group can experience symptomatic infection. Peripheral blood mononuclear cells (PBMCs) from study participants with enough material for the assessment of multiple endpoints ($n = 9$ symptomatic *P. vivax* malaria, 11 asymptomatic *P. vivax* malaria, and 12 healthy immune controls aged between 5 and 46) were selected for RNA-sequencing (RNA-seq) in the current study. No significant differences were detected between clinical groups within gender composition, hemoglobin levels, and platelet counts (Fig. 1A–C). Asymptomatic individuals were slightly younger than their symptomatic counterparts ($P = 0.0487$, Fig. 1D). As expected, parasite densities in this subset of samples were significantly higher in the symptomatic group compared with the asymptomatic group ($P = 0.0074$, Fig. 1E). The Duffy Binding Protein (DBP) antigen is a major *P. vivax* invasion ligand and an important target of naturally acquired immunity (Longley et al, 2016). Assessment of antibody responses to *P. vivax* DBP showed that all individuals in the healthy immune control group exhibited previous exposure to the *P. vivax* parasite (Fig. 1F). Importantly, clinical parameters and *P. vivax*-specific antibody responses in the RNA-seq study were representative of those previously reported in the mass cytometry cohort (Ioannidis et al, 2021) (Fig. EV1A–F).

## RNA-seq discriminates transcriptional profiles of symptomatic and asymptomatic *P. vivax* malaria and healthy immune controls

To identify molecular pathways underlying mechanisms of immunity to *P. vivax* malaria, PBMCs from *P. vivax*-infected symptomatic and asymptomatic study participants as well as healthy immune controls were profiled by RNA-seq. Multi-dimensional scaling of transcriptional profiles showed symptomatic *P. vivax* malaria to clearly differentiate from asymptomatic *P. vivax* malaria and healthy immune controls (Fig. 2A). After the detection and adjustment for known and unknown variation using the RUVr method (Risso et al, 2014) as described in the Structured Methods, clinical groups were incorporated as a factor into linear modeling for gene expression estimation and identification of differentially expressed genes. Differential expression analysis with pairwise comparisons between the three clinical groups revealed transcriptional changes within each contrast (Fig. 2B). Cell-type deconvolution predicted no significant differences between the frequencies of main PBMC cell types across clinical groups, suggesting that changes in transcriptional profiles were not associated to fluctuations in the composition of the blood mononuclear cell pool (Fig. 2C).

To gain biological insights into the immunological processes influencing differences in gene expression between clinical groups, we applied differential enrichment analysis using tmod (Zyla et al, 2019), with the significantly differentially expressed (DE) genes from pairwise comparisons between clinical groups as test sets, and the inbuilt blood transcriptional modules (Chaussabel et al, 2008; Li et al, 2013) as background gene sets (Fig. 2D). Numerous modules involved in cell division and cell cycle progression were enriched in symptomatic individuals compared with both asymptomatic participants and healthy immune controls. T-cell differentiation modules were upregulated in asymptomatic malaria compared to healthy immune controls. Interestingly, modules representative of various immune effector processes, including inflammation,

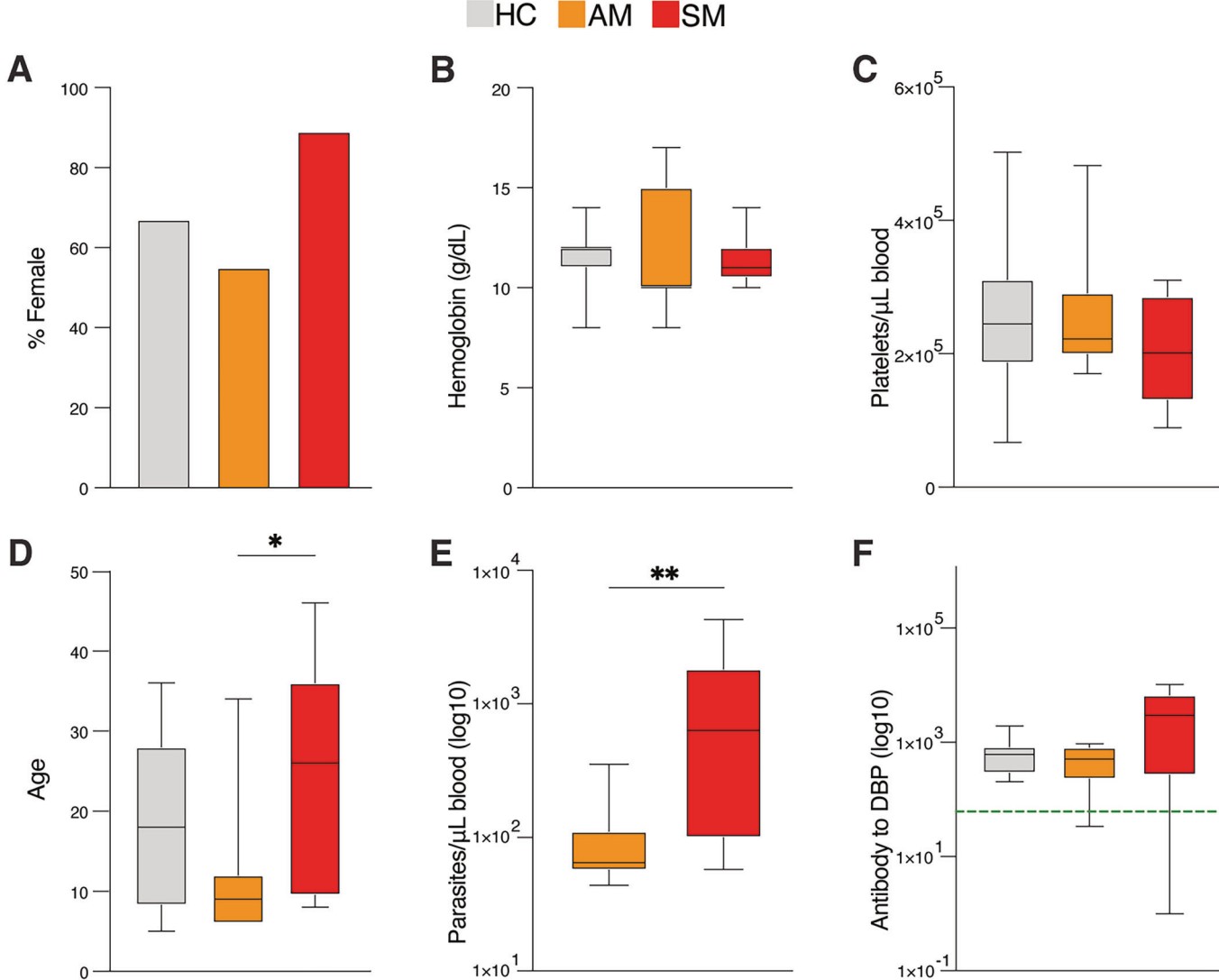

**Figure 1. Cohort study characteristics.**

PBMCs from *P. vivax* symptomatic ($n = 9$) and asymptomatic ($n = 11$) infected individuals as well as healthy immune controls ($n = 12$) were selected for RNA-seq analysis. **(A–F)** Clinical parameters determined in the study include gender **(A)**, hemoglobin (g/dL blood) **(B)**, platelets/µL blood **(C)**, age (*$P = 0.0487$) **(D)**, parasite density (**$P = 0.0074$) **(E)**, and IgG antibody to *P. vivax* Duffy binding protein **(F)**. Boxes represent the 25th to 75th percentile, whiskers show the range (min to max), and lines represent the median. Significance was determined by the Chi-square test **(A)** and the Kruskal–Wallis test with Dunn's multiple comparisons **(B–F)**.

antigen processing and presentation, toll-like receptor (TLR) signaling, inflammatory signaling, and monocytes were down-regulated in both symptomatic and asymptomatic malaria, suggesting *P. vivax* malaria infection of any severity and parasite density may impart detrimental effects upon monocyte function (Fig. 2D).

## Enhanced cell metabolism and T-cell differentiation discriminate between transcriptional profiles of symptomatic and asymptomatic *P. vivax* malaria

The vast majority of genes DE between symptomatic and asymptomatic *P. vivax*-infected individuals were uniquely regulated within this contrast (Fig. 3A). Unsupervised hierarchical clustering of expression profiles for all 442 genes DE between symptomatic

and asymptomatic malaria resulted in clear separation of transcriptional profiles across the two clinical groups (Fig. 3B). Gene-set enrichment using Ingenuity Pathway Analysis (IPA) canonical pathways, gene ontology (GO) terms and Kyoto Encyclopedia of Genes and Genomes (KEGG) pathways revealed upregulation of diverse metabolic processes including cellular metabolism, glycolysis, and phosphate metabolism in symptomatic *P. vivax* malaria (Figs. 3C,D and EV2). The gene *LACC1* encoding a purine enzyme with key roles in fatty-acid oxidation (Omarjee et al, 2021), short-chain dehydrogenase/reductase superfamily member *HSDL2* that regulates fatty-acid metabolism (Han et al, 2021) and the acyl coenzyme A synthetase *ACSF2* were upregulated within these pathways (Fig. 3D,E). Genes involved in ATP metabolism (*EHD1*), glycolysis (*GYG1*), and acetyl coenzyme A metabolism (*NAT1*) were also enriched in symptomatic and underrepresented

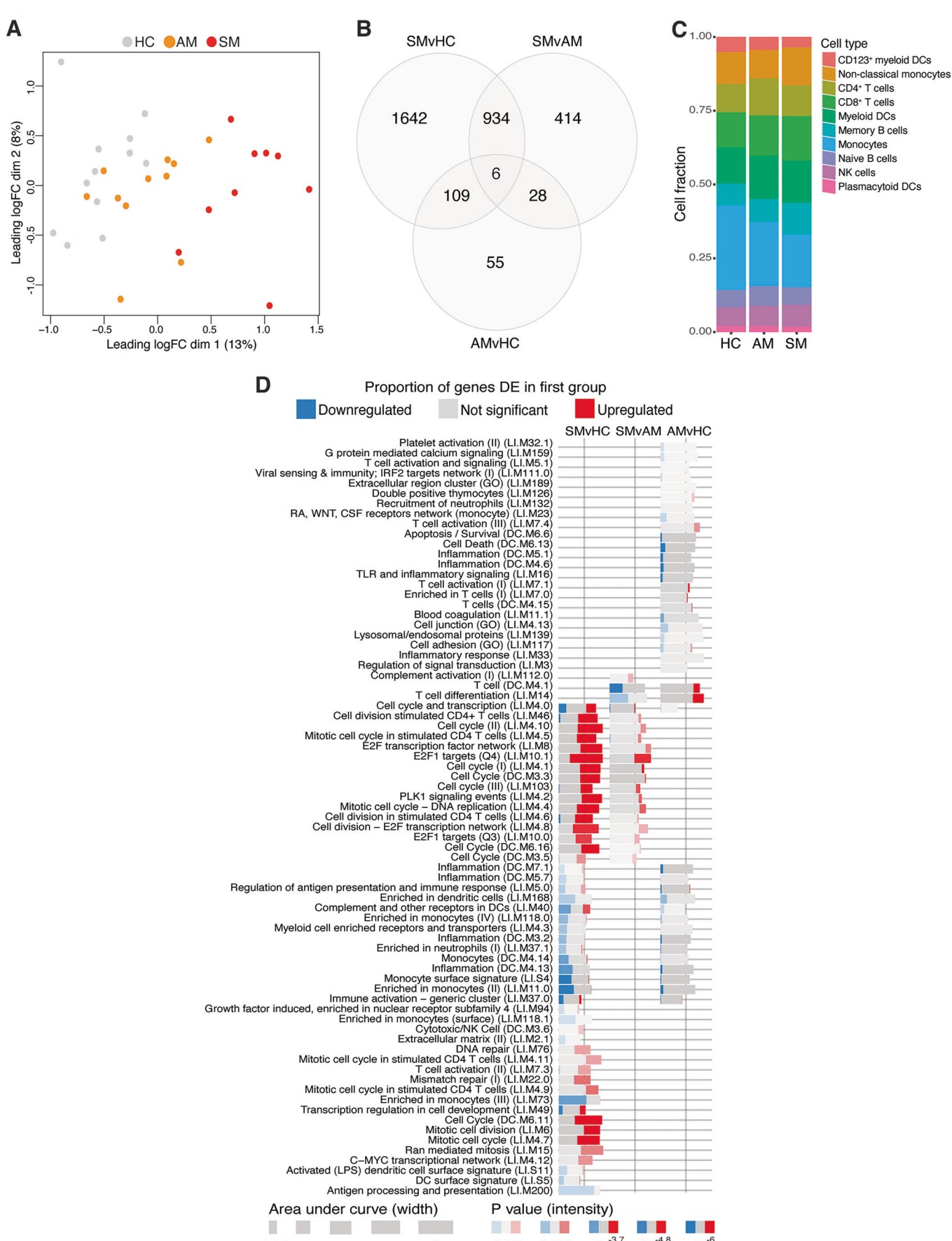

**Figure 2.    RNA-seq discriminates transcriptional profiles of *P. vivax* malaria and healthy immune controls.**

PBMCs from *P. vivax* symptomatic (*n* = 9) and asymptomatic (*n* = 11) infected individuals as well as healthy immune controls (*n* = 12) were selected for RNA-seq analysis. (A) Unsupervised multi-dimensional scaling plot displaying the top 500 most variably expressed genes across all samples. (B) Venn diagram displaying the number of differentially expressed genes determined at a false discovery rate (FDR) of 1% for pairwise comparisons between clinical groups. (C) Estimated proportions of PBMC populations determined from cell-type deconvolution of study participant transcriptional profiles. (D) Blood transcriptional module analysis showing significant modules differentially enriched for pairwise comparisons (FDR < 1%). Modules were identified with the tmodLimmaTest function, which implements Welch's *t* test.

in asymptomatic malaria (Fig. 3D,E). Pathway analysis also identified the upregulation of programmed cell death and proteasomal catabolic processes in symptomatic compared with asymptomatic *P. vivax* (Fig. 3C,D).

Gene ontology terms including somatic diversification of immune receptors and regulation of T-cell differentiation were enriched in individuals with asymptomatic relative to symptomatic *P. vivax* malaria (Fig. 3D). Genes mediating T-cell receptor (TCR) function were enriched in asymptomatic malaria, including *CAMK4, TESPA1,* and *CD5* that facilitate TCR enhancement, activation, and signaling, *LAT,* which encodes an enzyme phosphorylated following TCR activation (Zhang et al, 1999) and the transcription factors *TCF7* and *LEF1* involved in the regulation of peripheral T-cell differentiation via WNT signaling (Willinger et al, 2006) (Fig. 3E). Taken together, these data suggest that like previous findings in symptomatic *P. falciparum* malaria (Studniberg et al, 2022), symptomatic *P. vivax* infection might also pose an important metabolic burden on the host to control acute parasitemia, while asymptomatic *P. vivax* allows normal metabolic processes and favors T-cell differentiation.

## Symptomatic *P. vivax* malaria transcriptional profiles feature upregulation of checkpoint receptors and downregulation of monocyte effector processes

Most genes DE between symptomatic *P. vivax* malaria and healthy immune controls were uniquely regulated by this contrast (Fig. 4A). Unsupervised hierarchical clustering of all 2691 genes DE between symptomatic *P. vivax* malaria and healthy immune controls showed complete separation of transcriptional profiles (Fig. 4B). Consistent with Fig. 3, pathway enrichment with IPA canonical pathways, GO terms and KEGG pathways revealed upregulation of cell cycle, DNA damage, DNA repair, as well as pyruvate, phenylalanine, pyrimidine, and glycine metabolism in symptomatic *P. vivax* malaria (Figs. 4C,D and EV3). In addition, multiple genes involved in T-cell (*CD3E, ICOS*) and NK cell (*GZMA, KLRG1, SLAMF7*) effector function were upregulated in symptomatic individuals (Fig. 4D,E). Upstream regulator analysis predicted the master transcription factor *MYC,* which coordinates the transcriptional program for T-cell proliferation upon activation of the TCR-CD3 complex (Lindsten et al, 1988), along with *CD3E,* to be controlling these processes (Fig. 4F). Clinical *P. vivax* malaria also featured upregulation of pathways involved in cell death receptor signaling and IL-10 signaling (*PDCD1, PDCD2,* and *IL-10*), as well as numerous genes with immunoregulatory functions (Figs. 4C,D and EV3). Interestingly, the genes *CD160, EGR3, NOS3* with roles in the inhibition of T-cell function and previously found to be upregulated in response to asymptomatic *P. falciparum* malaria (Studniberg et al, 2022) were upregulated in *P. vivax* infections (Fig. 3D,E). Moreover, the immune checkpoint receptors *LAG3* and *CTLA4* were upregulated in symptomatic *P. vivax* malaria and present across multiple GO terms (Fig. 4E,F), suggesting that unlike

symptomatic *P. falciparum* malaria (Studniberg et al, 2022) that features a highly inflammatory response, clinical *P. vivax* malaria might drive the activation of feedback mechanisms to control a potentially excessive response to acute infection. In support of that proposition, comparison of transcriptional profiles of symptomatic *P. vivax*-infected individuals in this study with symptomatic *P. falciparum* malaria cases from the same endemic setting (Studniberg et al, 2022; Data ref: Studniberg et al, 2022) (Fig. EV4D–G) revealed numerous inflammatory GO terms involved in the stress response uniquely upregulated in response to clinical *P. falciparum* but not *P. vivax* malaria.

In addition to the upregulation of transcripts encoding protein products with anti-inflammatory function (*LAG3* and *CTLA4*), the downregulation of pro-inflammatory pathways was heavily featured by symptomatic *P. vivax* malaria, including *JAK-STAT* signaling, the T$_{H1}$ pathway, IL-1 signaling, and cytokine production (Figs. 4C,D and EV3). Many genes downregulated by symptomatic *P. vivax* malaria were associated with macrophage, monocyte and myeloid cell function, including genes with roles in antigen presentation (*HLA-DMA, HLA-DMB, HLA-DPA1, HLA-DPB1*), innate and pro-inflammatory immune responses (*IER3, IL24, IL23A, IL1R1, IRAK3, NLRP3*), chemokine receptors (*CCR1* and *CCR7*), TLR signaling (*TLR1, TLR2, TLR8*), and genes preferentially expressed in monocytes, (*CCL7, CD14, CD163, CD300E, CD300LB, CD302, CD74, CD93, FGR*) (Fig. 4D,E). Ingenuity upstream regulator analysis using the 1642 DE genes uniquely regulated by symptomatic *P. vivax* malaria compared with healthy immune controls (Fig. 4A) predicted among others, *CSF1,* a key cytokine facilitating the proliferation, survival and fate of monocytes and macrophages (MacDonald et al, 2010; Pierce et al, 1990), and *SPI1,* a positive regulator of monocyte differentiation (Rosa et al, 2007) to control the downregulation of these processes in symptomatic *P. vivax* malaria (Fig. 4F).

To infer cellular sources of symptomatic *P. vivax* malaria transcriptional profiles, cell-type-specific marker genes were examined using the dtangle package (Hunt et al, 2019) with haemopedia RNA-seq data as a reference (Choi et al, 2019) (Fig. 4G,H). Whilst transcriptional processes upregulated by symptomatic infection compared with healthy immune controls were mostly attributed to either T-cell or NK cell markers (Fig. 4G), 63% of marker genes downregulated by symptomatic malaria were predicted to originate from monocytes (Fig. 4H). Together, these results suggest that activation of feedback mechanisms to ameliorate excessive pro-inflammatory responses, as well as dysregulation of monocyte transcriptional activity, are features of symptomatic *P. vivax* malaria.

## *P. vivax* asymptomatic malaria supports T-cell differentiation but compromises the blood monocyte compartment

Figure 2D showed the downregulation of blood transcriptional modules associated with monocyte function in individuals with

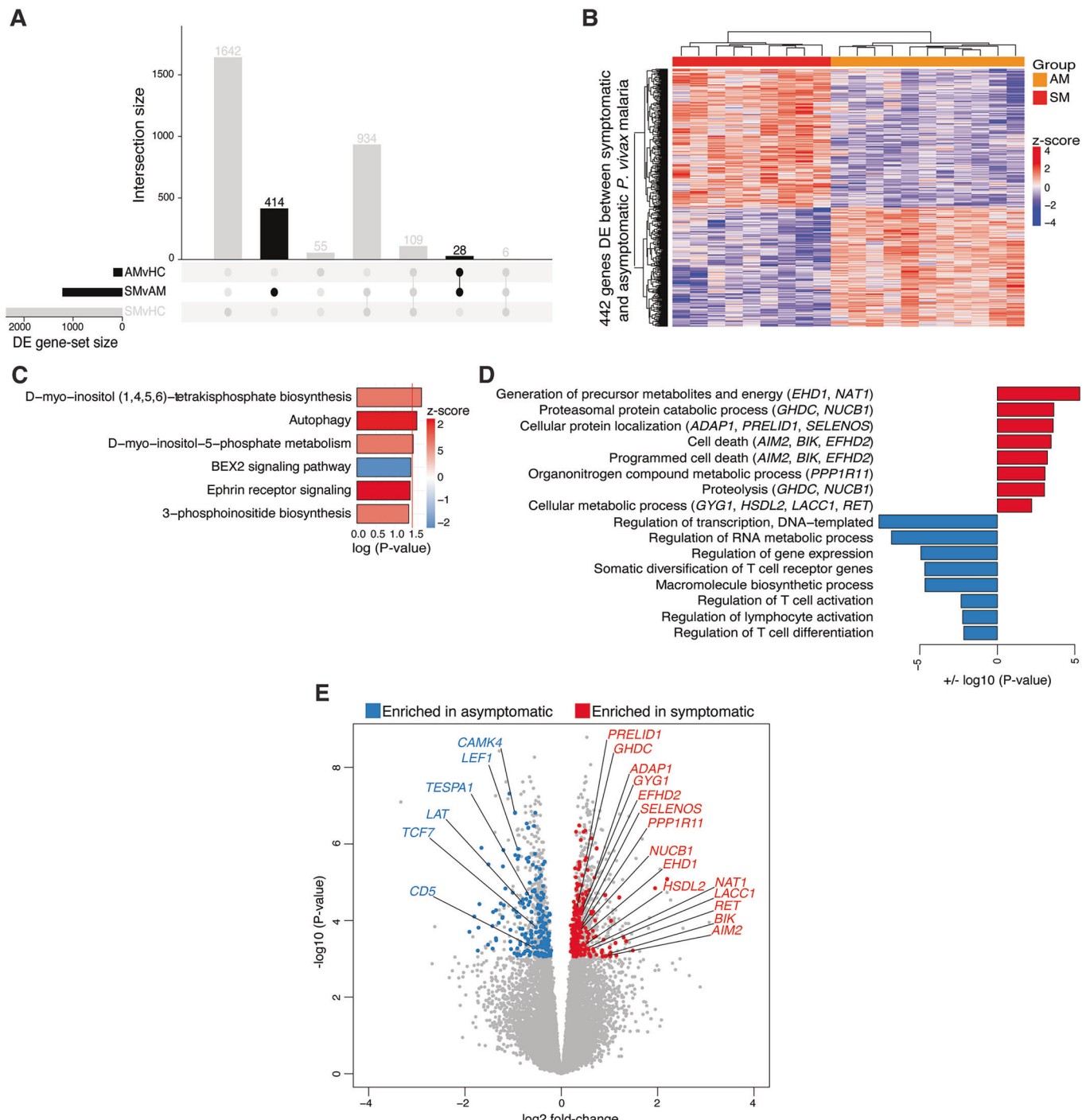

**Figure 3. Enhanced cell metabolism and T-cell differentiation discriminate between transcriptional profiles of symptomatic and asymptomatic *P. vivax* malaria.**

Gene expression profiles of PBMCs from symptomatic (SM, *n* = 9) and asymptomatic (AM, *n* = 11) *P. vivax*-infected participants were compared. (A) Unsupervised hierarchical clustering heatmap of the 442 differentially expressed (DE) genes between SM vs AM as well as in healthy immune controls (HC, *n* = 12). (B) UpSet plot showing the intersection between all DE genes in the experiment. Intersections for the 442 DE genes uniquely regulated by SM vs AM and shared with AM vs HC are in bold. (C) Bar plots showing significantly enriched gene ontology (GO) terms identified using hypergeometric testing implemented with the goana function in limma and scaled by + or −log10 (*P* value). Red GO terms are enriched in SM, and blue GO terms are enriched in AM. (D) Ingenuity canonical pathway analysis scaled by −log10 (*P* value) using the DE genes between SM vs AM and identified using a right-tailed Fisher's exact test. Pathways with a positive z-score in red are activated in SM, and pathways with a negative z-score in blue are activated in AM. The red line corresponds to a *P* value of 0.05. (E) Volcano plot displaying the 442 DE genes between SM vs AM *P. vivax* scaled by log2-fold change and −log10 (*P* value) obtained by the limma-voom differential expression analysis. Genes in red are overrepresented in SM and genes in blue are overrepresented in AM.

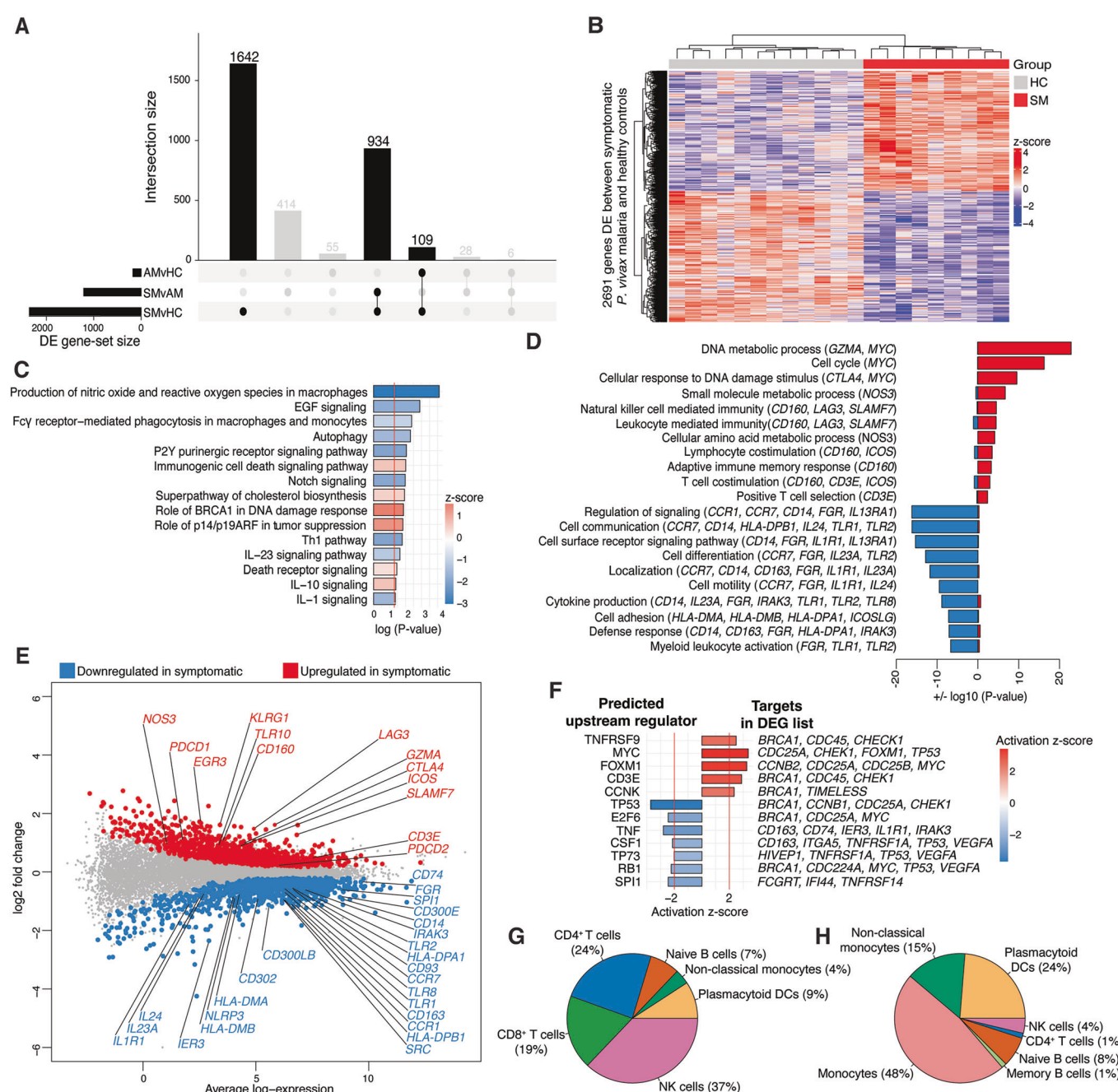

**Figure 4. Symptomatic *P. vivax* malaria transcriptional profiles feature upregulation of checkpoint receptors and downregulation of monocyte effector transcripts.**

Gene expression profiles of PBMCs from symptomatic (SM, n = 9) *P. vivax*-infected participants and healthy immune controls (HC, n = 12) were compared. (**A**) Unsupervised hierarchical clustering heatmap of the 2691 differentially expressed (DE) genes between SM vs HC. (**B**) UpSet plot showing the intersection between all DE genes in the experiment. Intersections for the 2691 DE genes regulated between SM vs HC are in bold. (**C**) Ingenuity canonical pathway analysis scaled by −log10 (*P* value) using the DE genes between SM vs HC and identified using a right-tailed Fisher's exact test. Pathways with a positive *z*-score in red are activated, and pathways with a negative *z*-score in blue are inhibited in SM compared with HC. The red line corresponds to a *P* value of 0.05. (**D**) Bar plots showing significantly enriched gene ontology (GO) terms identified using hypergeometric testing implemented with the goana function in limma and scaled by + or −log10 (*P* value). Red GO terms are upregulated, and blue GO terms are downregulated in SM. (**E**) Mean-difference plot displaying DE genes between SM vs HC. Each gene is plotted as a single point determined by log2-fold change and average transcript abundance. Red genes are overrepresented, and blue genes are underrepresented in SM. (**F**) Ingenuity upstream regulator analysis using the DE genes between SM vs HC. Predicted upstream regulators with a *z*-score >2 are activated and predicted upstream regulators with a *z*-score <2 are inhibited in SM compared with HC. The red line corresponds to a *z*-score activation of +/− 2. (**G, H**) Estimated proportions of PBMC subpopulations determined from cell-type deconvolution featuring transcriptional profiles upregulated (**G**) and downregulated (**H**) in SM compared with HC.

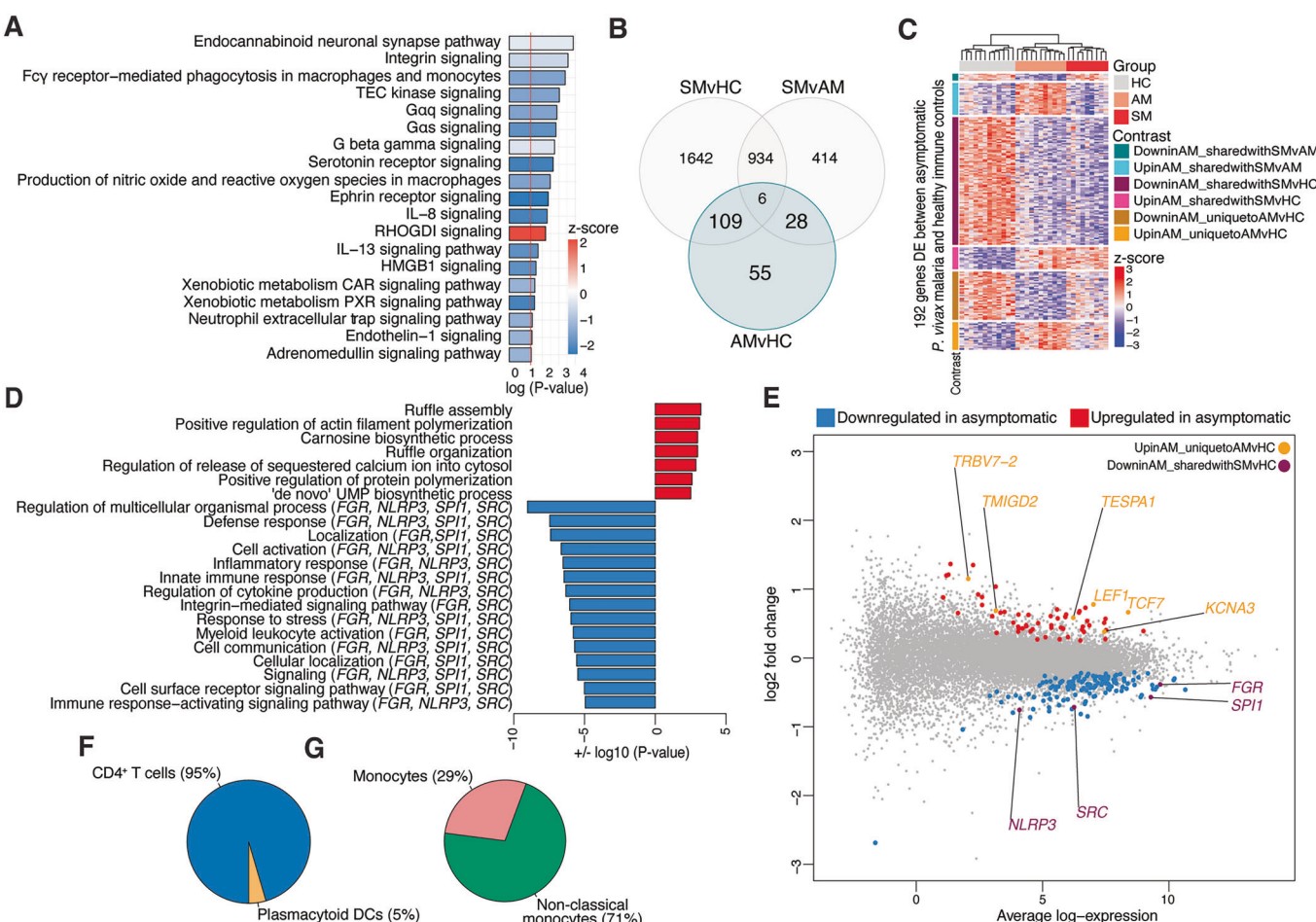

**Figure 5.** *P. vivax* **asymptomatic malaria features transcriptional profiles supporting T-cell differentiation but compromising blood monocyte function.**

Gene expression profiles of PBMCs from asymptomatic (AM, $n = 11$) *P. vivax*-infected participants and healthy immune controls (HC, $n = 12$) were compared. (A) Ingenuity canonical pathway analysis scaled by −log10 (*P* value) using the DE genes between AM vs HC and identified using a right-tailed Fisher's exact test. Pathways with a positive *z*-score in red are activated, and pathways with a negative *z*-score in blue are inhibited in AM compared with HC. The red line corresponds to a *P* value of 0.05. (B) Venn diagram showing the number of differentially expressed (DE) genes determined at a false discovery rate of 1% for pairwise comparisons between clinical groups. The 192 genes DE between *P. vivax* AM vs HC and co-regulated by the SM vs HC and SM vs AM pairwise comparisons are bolded and shaded in blue. (C) Heatmap displaying the 192 genes DE between *P. vivax* AM vs HC and co-regulated by the SM vs HC or SM vs AM pairwise comparisons. (D) Bar plots showing significantly enriched gene ontology (GO) terms for the 109 DE genes co-regulated between the AM vs HC and SM vs HC pairwise comparisons identified using hypergeometric testing implemented with the goana function in limma and scaled by + or −log10 (*P* value). Red GO terms are upregulated, and blue GO terms are downregulated in AM. (E) Mean-difference plot displaying DE genes between AM vs HC. Each gene is plotted as a single point determined by log2-fold change and average transcript abundance. Red genes are overrepresented, and blue genes are underrepresented in AM. Genes of interest that are uniquely upregulated by AM are in yellow. Genes of interest that are downregulated by both AM and SM are in maroon. (F, G) Estimated proportions of PBMC subpopulations determined from cell-type deconvolution featuring transcriptional profiles from the 83 genes upregulated (F) and downregulated (G) by AM.

asymptomatic *P. vivax* malaria in addition to those with symptomatic infection. To further explore transcriptional processes that may be influencing monocyte function during asymptomatic *P. vivax* malaria, we investigated gene expression changes between individuals with asymptomatic *P. vivax* malaria and healthy immune controls. Ingenuity pathway enrichment showed significant downregulation of numerous signaling pathways, Fcγ receptor-mediated phagocytosis, and the monocyte pro-inflammatory cytokine interleukin-8 (IL-8) (Meniailo et al, 2018) in asymptomatic *P. vivax* malaria compared to healthy immune controls (Fig. 5A). Of the 198 genes differentially regulated by asymptomatic *P. vivax*, 109 were co-regulated with symptomatic *P. vivax* malaria (Fig. 5B,C). Gene ontology and KEGG pathway

analysis with this gene set revealed downregulation of pathways associated with inflammation and monocyte function, including the innate immune response, regulation of cytokine production, myeloid leukocyte activation, and phagosome pathways (Figs. 5D and EV5). Importantly, many genes downregulated by symptomatic *P. vivax* malaria in Fig. 4D,E, including *SPI1*, implicated in monocytic autophagy (Xie et al, 2023) and *FGR*, encoding a monocytes tyrosine kinase (Hatakeyama et al, 1996) with pro-inflammatory action (Acin-Perez et al, 2020), were also downregulated during asymptomatic *P. vivax* infections (Fig. 5D,E), suggesting that *P. vivax* malaria infection of any parasite density may influence monocyte function. To provide proof of concept for that proposition, isolated CD14+ monocytes from symptomatic and

asymptomatic infected individuals as well as healthy immune controls were stimulated with LPS and their capacity to secrete cytokines was determined. Whereas IL-1β secretion was significantly reduced by monocytes from symptomatic individuals compared to healthy immune controls, TNF production was significantly compromised in monocytes from symptomatic *P. vivax* individuals relative to asymptomatic infected individuals and healthy immune controls (Fig. EV6).

Gene ontology enrichment with the 83 genes uniquely regulated by asymptomatic malaria compared with healthy immune controls (Fig. 5B) was performed to identify processes exclusively regulated by asymptomatic *P. vivax* infection. Consistent with Figs. 4 and 5, somatic diversification of TCR genes, T-cell differentiation and T-cell activation were uniquely upregulated by asymptomatic malaria (Fig. EV7A). Upregulated genes enriched within these terms included *KCNA3*, encoding a potassium channel with roles in TCR signaling (Feske et al, 2015), and the costimulatory molecule *TMIGD2*, expressed in response to TCR signaling to facilitate T-cell proliferation and cytokine production (Zhu et al, 2013) (Fig. EV7A,B). Whereas transcriptional profiles upregulated by asymptomatic *P. vivax* malaria were predominantly associated with canonical CD4$^+$ T-cell markers (95%), 100% of marker genes downregulated by asymptomatic infection were predicted to arise from monocytes (Fig. 5F,G). Together, these data support the concept that while *P. vivax* asymptomatic malaria can facilitate T-cell differentiation and function, the blood monocyte compartment appears dysregulated by these low parasitemia infections.

## Weighted gene co-expression network analysis of the immune response to *P. vivax* malaria

Applying high-dimensional mass cytometry to the cohort described in this study, we previously identified discrete populations of CD4$^+$ T cells, T$_{FH}$ cells, and memory B cells (MBCs) associated with reduced or increased risk of symptomatic and asymptomatic *P. vivax* malaria (Ioannidis et al, 2021). Our main findings revealed that whereas high frequencies of class-switched but not IgM$^+$ MBCs predicted reduced odds of symptomatic *P. vivax* malaria, populations of T$_{H}1$ cells with a stem central memory phenotype and T$_{H}2$ polarized T-regulatory (T$_{REG}$) cells reduced risk of asymptomatic *P. vivax* malaria, suggesting that activation of cell-mediated immunity might be required to control low-density asymptomatic *P. vivax* infections. A powerful method to identify the relationship of key biological features with whole-genome expression profiles without losing genes to fold-change thresholds imposed by differential expression pipelines is through weighted co-expression network analysis, (Langfelder and Horvath, 2008; Zhang and Horvath, 2005). To identify associations between transcriptional profiles, clinical traits, inferred monocyte frequencies, and populations of T cells and B cells with increased or reduced risk of *P. vivax* malaria (Ioannidis et al, 2021), a signed co-expression network was constructed in WGCNA using the limma-voom transformed expression data, a soft-thresholding power of 18, and a minimum cluster size of 30. Modules with similar expression were merged using a dynamic tree cut height of 0.35, resulting in a total of 15 gene co-expression modules (Fig. 6A). To identify modules containing genes significantly enriched in our DE analysis, we performed multiple rotational gene-set testing using the mroast function in limma, using module genes as test sets against the

differential expression results for each contrast. Our analysis identified six-gene modules with significant enrichment for genes either upregulated or downregulated in at least one pairwise comparison between clinical groups (Tables EV1–3). These modules were selected to explore correlations with clinical variables (Figs. 6B and EV8). Symptomatic *P. vivax* malaria correlated significantly with modules black and greenyellow (Fig. 6B). Hierarchical clustering of genes within these modules that were also DE in the limma-voom analysis revealed a transcriptional response predominately upregulated in symptomatic *P. vivax* malaria compared with both the asymptomatic malaria and healthy immune control groups (Fig. 6C,D). While module black significantly correlated with parasitemia (Fig. 6B), and featured genes enriched for biological processes mostly involved in metabolism and protein transport (Table 1), module greenyellow genes were enriched for terms involved in cell cycle, immunoglobulin production, adaptive immune response, cellular response to stress, and B-cell receptor signaling (Table 1).

Differentially expressed genes featuring within modules grey60, pink, and salmon were significantly upregulated in the healthy immune control group compared with both asymptomatic and symptomatic *P. vivax* malaria (Fig. 6E–G). GO terms enriched for module grey genes were involved in the humoral immune response and myeloid cell development and differentiation, while GO terms enriched for module pink genes were associated with myeloid cell activation and differentiation, as well as IL-1, IL-8, and IL-12 production—all cytokines with major roles in myeloid-derived cellular immunity (Table 1). Visualization of DE genes featured in the tan module showed that most of these genes were upregulated by asymptomatic *P. vivax* malaria (Fig. 6H). Noticeably, GO terms enriched for all tan module genes included diversification of TCR genes, T-cell differentiation, and TCR recombination (Table 1), consistent with our results in Figs. 3 and 5.

To build a comprehensive picture of the immune response to *P. vivax* malaria in an endemic setting, a multiscale model was constructed using raw Spearman correlation coefficients calculated between gene modules, module hub genes, cell populations associated with reduced or increased risk of *P. vivax* malaria (Ioannidis et al, 2021), inferred monocyte frequencies (Fig. 2C), clinical variables, and clinical groups (Fig. 6I). Under prefuse force-directed layout based on correlation strength (Heer et al, 2005), a well-organized cluster formed around parasitemia levels, symptomatic *P. vivax* malaria, and the greenyellow module. Positive associations were found between module greenyellow and multiple cell frequencies associated with recent exposure to symptomatic *P. vivax* malaria, including T-bet$^+$ atypical and activated MBCs, and CXCR3$^+$PD-1$^+$T-bet$^+$ T$_{FH}$ cells predictive of increased risk of symptomatic *P. vivax* infection (Fig. 6I). Thus, this network supports a model by which high parasitemia levels during symptomatic infection drive a proliferative transcriptional profile supporting the expansion of T-bet$^+$ MBCs and T$_{FH}$ cells to help control acute infection.

Our prefuse force-directed layout allowed for visualization of a cluster featuring transcriptional profiles rich in terms supporting cell metabolism (salmon module) and humoral immunity (grey module), positively associating with the healthy immune control group, class-switched MBCs, as well as T$_{H}2$ polarized T$_{FH}$ cells and T$_{REG}$ cells with strong associations with reduced odds of symptomatic *P. vivax* malaria (Fig. 6I). Within this cluster,

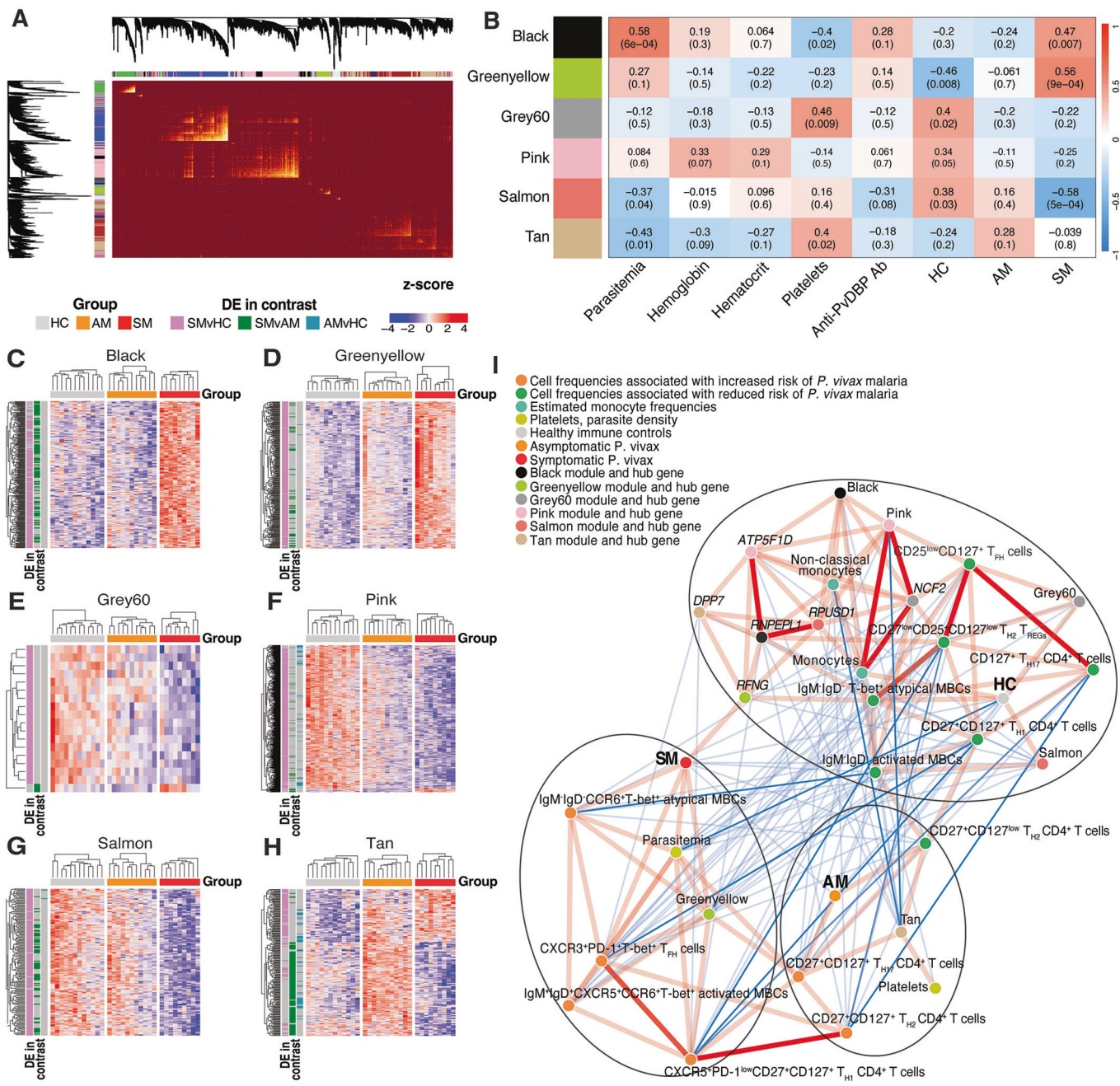

**Figure 6.  Weighted gene co-expression network analysis of the immune response to *P. vivax* malaria.**

Co-expression network analysis with whole-genome expression profiles of PBMCs from *P. vivax* symptomatic (SM, $n = 9$), asymptomatic (AM, $n = 11$) and healthy immune control (HC, $n = 12$) individuals. (A) Network visualization of all genes in the analysis using a heatmap of the topological overlap matrix. Colors in the bars represent gene co-expression modules ($n = 15$). (B) Heatmap depicting correlations between gene co-expression modules significantly enriched for at least one pairwise contrast in the limma-voom differential expression analysis, clinical traits, and clinical groups. Red represents positive correlations and blue represents negative correlations. (C–H) Semi-supervised heatmaps showing expression of genes differentially expressed in the limma-voom analysis, and identified within co-expression modules Black (C), Greenyellow (D), Grey60 (E), Pink (F), Salmon (G), and Tan (H). (I) Spearman correlations between gene co-expression modules, module eigengenes, cell populations associated with either increased or reduced risk of *P. vivax* malaria (Ioannidis et al, 2021), estimated monocyte frequencies, clinical traits, and clinical groups. Correlations were visualized in a network created with the MetScape plugin in Cytoscape using a prefuse force-directed layout based on correlation strength. Line color transparency and width represent correlation strength, with wide opaque lines representing stronger correlations, and narrow transparent lines representing weaker correlations. Red lines represent positive correlations, and blue lines represent negative correlations. A minimum correlation cut-off of 0.35 was employed for visualization.

**Table 1. Gene ontology enrichment analysis using WGCNA module genes.**

| Module (no. of genes) | Gene Ontology Term[a] | Overlap | P value |
|---|---|---|---|
| Black (868) | Nucleoside phosphate metabolic process | 64/505 | 3.17E-10 |
| | Purine nucleoside metabolic process | 58/447 | 9.03E-10 |
| | ATP metabolic process | 29/192 | 6.94E-07 |
| | Protein transport | 104/1335 | 1.5987E-04 |
| | Toxin transport | 7/32 | 1.4923E-03 |
| Greenyellow (823) | Cell cycle process | 188/1086 | 3.08E-52 |
| | Immunoglobulin production | 75/180 | 1.09E-48 |
| | Adaptive immune response | 102/560 | 1.38E-29 |
| | Cellular response to stress | 143/1679 | 1.34E-09 |
| | B-cell receptor signaling pathway | 12/71 | 2.79E-04 |
| Grey60 (317) | Antimicrobial humoral response | 12/52 | 3.53E-10 |
| | Myeloid cell development | 12/75 | 2.96E-08 |
| | Myeloid cell differentiation | 22/350 | 2.46E-06 |
| | Humoral immune response | 13/135 | 3.30E-06 |
| | Myeloid cell homeostasis | 10/153 | 1.00E-03 |
| Pink (3083) | Myeloid leukocyte activation | 87/188 | 6.62E-17 |
| | Interleukin-8 production | 34/65 | 3.62E-09 |
| | Interleukin-1 beta production | 39/80 | 3.71E-09 |
| | Myeloid leukocyte differentiation | 65/180 | 1.33E-07 |
| | Interleukin-12 production | 22/51 | 9.90E-05 |
| Salmon (490) | Primary metabolic process | 300/7736 | 2.39E-08 |
| | Protein modification process | 136/2936 | 3.18E-07 |
| | Protein ubiquitination | 44/722 | 1.54E-05 |
| | Steroid biosynthetic process | 12/110 | 1.56E-04 |
| | Autophagy | 29/510 | 1.00E-03 |
| Tan (2003) | Adaptive immune response | 122/560 | 7.66E-10 |
| | T-cell differentiation in the thymus | 23/76 | 4.28E-05 |
| | Gamma-delta T-cell activation | 9/24 | 1.83E-03 |
| | T-cell receptor V(D)J recombination | 3/4 | 7.39E-03 |
| | T-cell differentiation | 48/269 | 8.85E-03 |

[a]Genes from six WGCNA gene modules with significant enrichment in at least one pairwise comparison from the limma-voom differential expression pipeline were entered into gene ontology pathway analysis.

predicted monocyte frequencies formed strong associations with the black and pink modules. Whereas only non-classical monocytes were positively correlated with symptomatic *P. vivax* malaria, frequencies of classical monocytes and the transcriptional profile supporting myeloid cell function (pink module), were only positively associated with healthy immune controls (Fig. 6I), suggesting that a functional monocyte compartment might play an important role in efficient control of parasitemia and immunity to *P. vivax* malaria.

Transcriptional profiles supporting T-cell differentiation (tan module) clustered together with CD27$^+$CD127$^+$ $T_{H2}$ and CD27$^+$CD127$^+$ $T_{H17}$ polarized cells associated with reduced and increased risk of symptomatic *P. vivax* malaria, respectively (Fig. 6I). Thus, the transcriptional profile supporting the induction of T-cell responses driven by asymptomatic *P. vivax* malaria might not be uniquely associated with cellular responses involved in the control of parasitemia. Moreover, consistent with our findings in Figs. 2, 4, and 5, predicted monocyte frequencies and gene modules (pink) supporting monocyte function remained negatively associated with both the symptomatic and asymptomatic clinical groups, suggesting that monocyte dysregulation might be a feature of *P. vivax* malaria.

## Downregulation of specific monocyte transcriptional programs associated with symptomatic and asymptomatic *P. vivax* malaria

To identify genes and myeloid cell processes specifically compromised by symptomatic or asymptomatic *P. vivax* malaria, we used elastic net regularized logistic regression modeling. Genes enriched across monocyte blood transcriptional modules, gene targets of predicted upstream regulators *CSF1* and *SPI1*, and myeloid cell-associated GO terms identified in Figs. 4 and 5 were selected. We then further restricted this gene set to those co-expressed within the pink WGCNA module. Genes were then classified as being downregulated by symptomatic *P. vivax* malaria or by both symptomatic and asymptomatic infection and used as input for logistic modeling. Leave-one-out cross-validation variable selection revealed a six-gene model (*APP, HLA-DMA, HLA-DPA1, IER3, MAP3K1, P2RY2*) downregulated by symptomatic *P. vivax* that correctly classified *P. vivax* malaria from healthy immune controls (AUC = 0.78, Fig. 7A–C). Common features between genes selected from model classification were identified using functional association data (Montojo et al, 2010) and visualized using a prefuse force-directed layout in Cytoscape (Shannon et al, 2003) (Fig. 7D). Model genes downregulated by symptomatic *P. vivax* malaria appeared to be involved in critical myeloid cell pathways including antigen presentation and signaling cascades leading to cytokine secretion, cell mobilization, and control of cell survival. Specifically, *HLA-DMA* and *HLA-DPA1* encode human leukocyte antigen alpha chain paralogues—components of the class II major histocompatibility complex (MHC), essential for antigen presentation. Whereas *MAP3K1* and *IER3* encode the mitogen-activated protein kinase 1 and the immediate early response 3 protein, respectively, with important roles in the control of cell survival and induction of inflammatory responses (Arlt and Schafer, 2011; Pham et al, 2013; Suddason and Gallagher, 2015), activation of amyloid precursor protein (*APP*) and purinergic P2 receptors (*P2RY2*) potentiates TLR-induced production of pro-inflammatory cytokines (Alberto et al, 2022; Eckfeld et al, 2023; Spitzer et al, 2020).

From the genes downregulated by both symptomatic and asymptomatic *P. vivax* malaria, a two-gene model discriminated *P. vivax* malaria from healthy immune controls (AUC = 0.75,

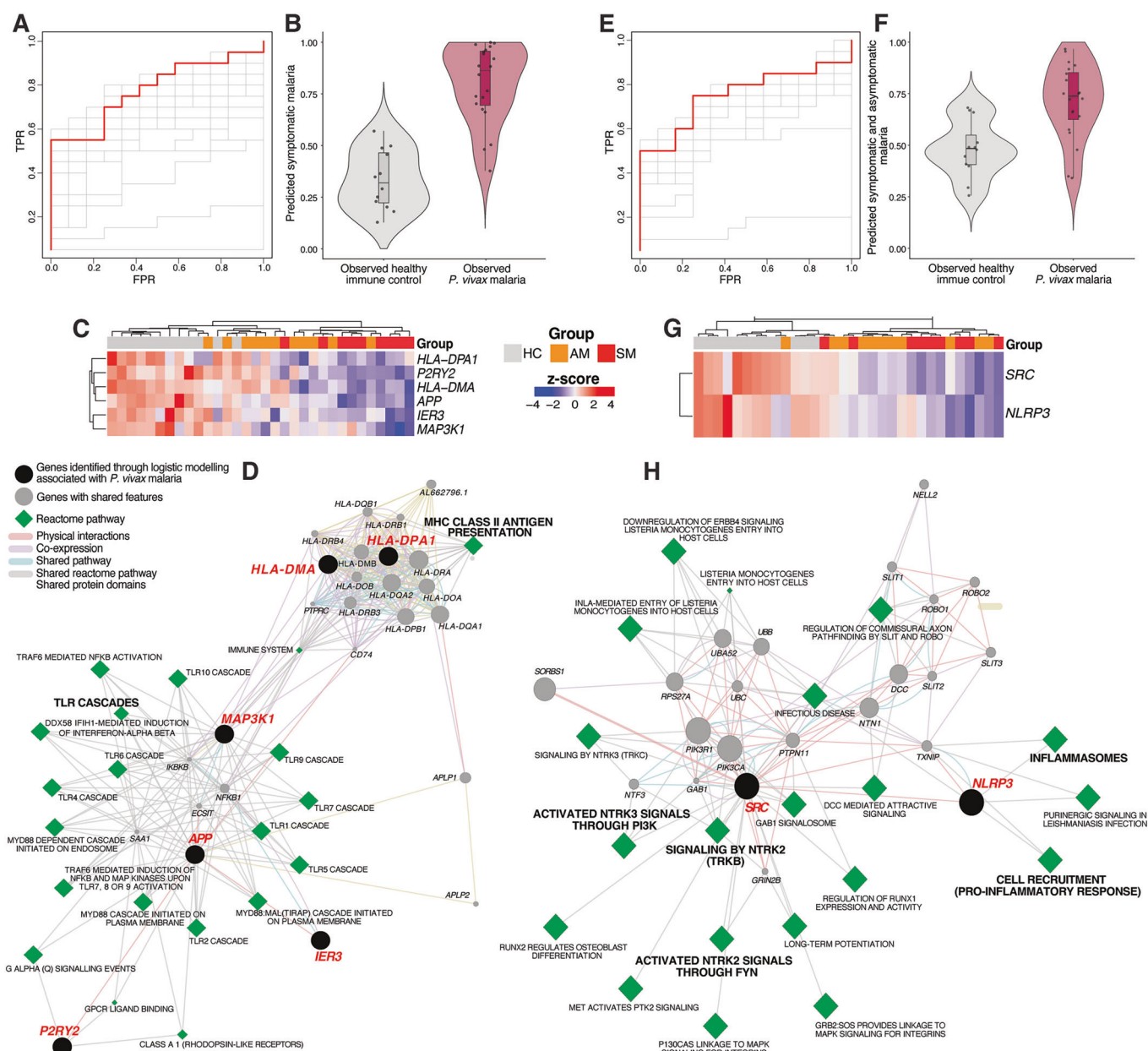

**Figure 7. Downregulation of specific monocyte transcriptional programs associated with symptomatic and asymptomatic *P. vivax* malaria.**

Genes present in the pink WGCNA module and enriched across monocyte blood transcriptional modules, gene targets of predicted upstream regulators *CSF1* and *SPI1*, and myeloid cell-associated GO terms identified in Figs. 4 and 5 were submitted for logistic modeling. (**A, B**) ROC curve (**A**) and predicted probability plot (**B**) classifying individuals infected with *P. vivax* malaria (*n* = 20) or healthy immune controls (*n* = 12) based on gene expression of DE genes meeting the above criteria and regulated by symptomatic *P. vivax* malaria against healthy immune controls (*n* = 148 genes). (**C**) Unsupervised hierarchical clustering heatmap displaying gene expression for the six genes selected by the model to be predictive of *P. vivax* malaria. (**D**) Genes and Reactome pathways sharing features with the six-gene predictive model. (**E, F**) ROC curve (**E**) and predicted probability plot (**F**) classifying individuals infected with *P. vivax* malaria or healthy immune controls based on gene expression of DE genes meeting the above criteria and regulated by symptomatic and asymptomatic *P. vivax* malaria against healthy immune controls (*n* = 23 genes). (**G**) Unsupervised hierarchical clustering heatmap of gene expression for the two genes selected by the model to be predictive of *P. vivax* malaria. (**H**) Genes and reactome pathways sharing features with the two-gene predictive model. Box plots represent the 25th to 75th percentile, whiskers extend to the furthest data points within 1.5× the interquartile range from the hinges, and lines represent the median.

Fig. 7E–G), identifying myeloid cell processes still downregulated by low parasitemia asymptomatic infection. These included *SRC*, a non-receptor tyrosine kinase with roles in monocyte chemotaxis (Kumar et al, 2001) and intracellular signaling leading to cytokine and chemokine secretion (Selvaraj et al, 2003), and *NLRP3*, a

subunit of the inflammasome, implicated in monocyte pro-inflammatory cytokine secretion (Gaidt et al, 2016) (Fig. 7G,H). Together, these results are consistent with the notion that while high parasitemia symptomatic *P. vivax* infection might induce generalized downregulation of monocyte transcriptional activity,

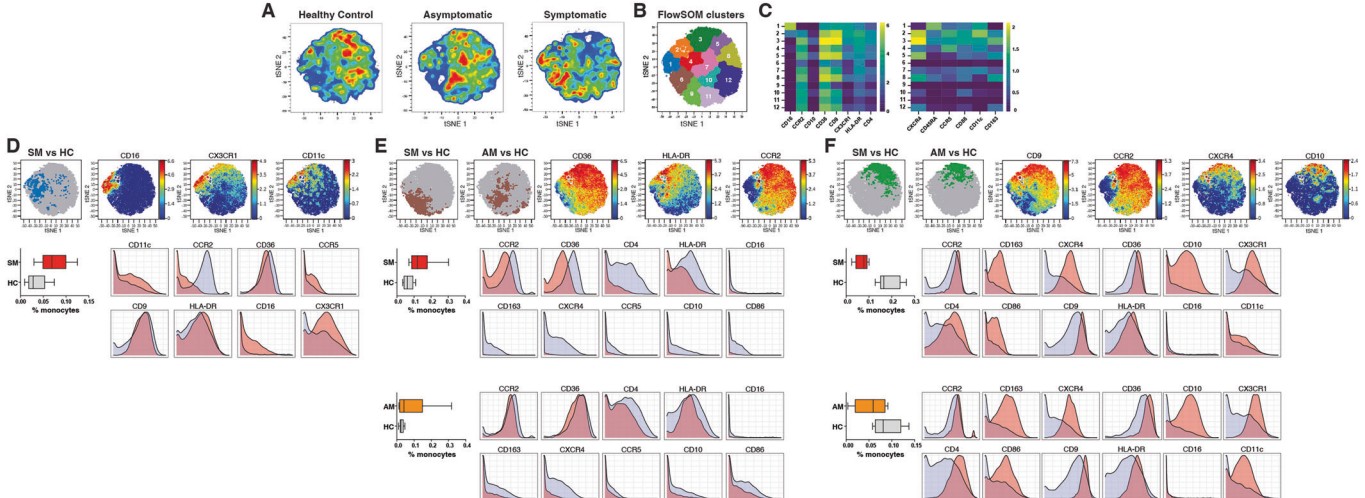

**Figure 8. Identification of monocyte subpopulations induced by *P. vivax* symptomatic and asymptomatic malaria.**

(A–F) PBMCs from *P. vivax* symptomatic ($n = 8$) and asymptomatic ($n = 8$) infected individuals as well as healthy immune controls ($n = 8$) were stained with a panel of metal-labeled antibodies and analyzed by CyTOF. (A) t-SNE analysis was performed on CD14+ CD3-CD29- monocytes. Plots display cell density and represent the pooled data for each group. (B) Projection of FlowSOM clusters on a t-SNE plot. (C) Heatmaps show the median marker expression for each FlowSOM cluster. (D–F) Unsupervised identification of differentially abundant populations between clinical groups was performed using CITRUS (FDR < 5%) between clinical groups. Panels on the top left or each panel show differentially abundant populations on a t-SNE overlay identified as follows: significantly more abundant in symptomatic individuals compared to healthy immune controls (D), significantly more abundant in symptomatic and asymptomatic individuals compared to healthy immune controls (E) and significantly depleted in symptomatic and asymptomatic individuals compared to healthy immune controls (F). vi-SNE plots on the top right-hand side or each top panel depict relevant marker expression on t-SNE overlays. The lower left panels show the frequency of differentially abundant cell populations identified by CITRUS. Boxes represent the 25th to 75th percentile, whiskers show the range (minimum to maximum), and lines represent the median of 8 symptomatic, 8 asymptomatic, and 8 healthy immune control biological replicates. The lower right panels illustrate marker expression in differentially abundant populations (pink histograms), relative to background expression (lilac histograms).

compromising cell activation and antigen presentation, asymptomatic *P. vivax* malaria might be sufficient to have a detrimental impact on the ability of the host to support adequate inflammatory function.

## High-dimensional mass cytometry identifies a depletion of blood CD10+CCR2+CXCR4+ classical monocytes in symptomatic and asymptomatic *P. vivax* malaria

Figures 2–7 revealed significant downregulation of transcriptional pathways involved in monocyte function during both symptomatic and asymptomatic *P. vivax* malaria. To determine if those transcriptional signatures were associated with changes in the composition of the blood monocyte pool, we designed a panel of 20 metal-labeled antibodies specific for a range of known and putative monocyte markers (Fig. EV9). PBMCs from symptomatic and asymptomatic *P. vivax*-infected individuals as well as healthy immune controls were then stained with metal-labeled antibodies and analyzed by mass cytometry. To explore the composition of the monocyte compartment, t-SNE analysis (Fig. 8A) and FlowSOM clustering (Fig. 8B) were performed on gated CD3− CD19− CD14+ cells (Fig. EV9), and marker expression was assessed in each cell population (Fig. 8C). This approach allowed for the identification of 12 distinct subpopulations of monocytes including, 10 classical (CD14+CD16−), 1 intermediate (CD14+CD16low) and 1 non-classical (CD14+CD16high) monocyte sub-population (Fig. 8A–C). All monocyte subsets expressed CD36 and CD45. Most, but not all, monocyte subpopulations expressed

CCR2. Expression of CX3CR1, HLA-DR, and CD4 was either low or absent in most subsets.

Unsupervised identification of differentially abundant cell populations between clinical groups was performed using the CITRUS algorithm (FDR < 5%) (Bruggner et al, 2014). This analysis revealed a population of CD16+CX3CR1+CD11c+CCR2low non-classical monocytes (resembling cluster 1 in the FlowSOM analysis) that was significantly increased in individuals with symptomatic *P. vivax* malaria compared with healthy immune controls (Fig. 8D). Subset of CD36+ classical monocytes (resembling cluster 6 in the FlowSOM analysis) were also significantly more abundant not only in symptomatic but also asymptomatic *P. vivax*-infected individuals relative to controls (Fig. 8E). In contrast, subsets of CD10+CCR2+CXCR4high classical monocytes (resembling cluster 3 in the FlowSOM analysis) were significantly reduced in both *P. vivax* malaria symptomatic and asymptomatic individuals relative to healthy immune controls. (Fig. 8F). Thus, together these findings suggest that the reduced monocyte transcriptional activity observed in *P. vivax* malaria might be associated with a depletion of a subset of classical monocytes expressing high levels of chemokine receptors from peripheral blood.

## Discussion

Compared with *P. falciparum*, host transcriptional profiles of *P. vivax*-infected individuals have not been extensively investigated. Our systems immunology approach integrating transcriptional profiling, mass cytometry cellular phenotypes, and clinical

parameters of *P. vivax*-infected individuals allowed to untangle the complexity of mechanisms underlying naturally acquired immunity to *P. vivax* malaria as well as immunoregulatory mechanisms operating at different parasitic load infections.

This study provided a rare opportunity to compare transcriptional profiles of *P. falciparum* and *P. vivax* symptomatic malaria from cohorts derived from within the same endemic setting and conducted at the same time (Studniberg et al, 2022). Unlike symptomatic *P. falciparum* malaria that drives strong inflammatory blood transcriptional profiles (Loughland et al, 2020; Studniberg et al, 2022; Tran et al, 2016), our results revealed that many pro-inflammatory pathways, including cytokines and their receptors, as well signaling pathways were downregulated by acute symptomatic *P. vivax* malaria. Furthermore, despite increased metabolic rates, proliferation, and expansion of discrete populations of B cells and T cells, symptomatic *P. vivax* malaria featured a tolerogenic profile, with the upregulation of numerous genes involved in the downregulation of T-cell function, including *EGR2, EGR3, DHRS9, LAG3*, and *CTLA4*. This immunoregulatory transcriptional signature observed in response to symptomatic *P. vivax* malaria is significantly aligned with various genes involved in the negative regulation of T-cell function previously found to be upregulated in asymptomatic *P. falciparum* malaria carriers, with the inhibitory receptor CTLA4 predicted to control many of these processes (Studniberg et al, 2022). Similarly, elevated expression of the immunoregulatory molecules PD-1, and TIM-3 has been reported in response to acute *P. vivax* infection (Costa et al, 2015) and increased frequencies of FoxP3$^+$ T-regulatory cells expressing CTLA4 were observed in Brazilian adults with symptomatic *P. vivax* malaria (Ferreira et al, 2023). Consistent with these findings in naturally acquired infections, transcriptional profiles from controlled human malaria infections have been found to feature upregulation of genes with immunosuppressive function at the onset of blood-stage *P. vivax* malaria (Rojas-Pena et al, 2018) (Vallejo et al, 2018). Altogether, this evidence supports the concept that these immunoregulatory mechanisms might become activated to mitigate potentially pathogenic inflammatory responses elicited early in response to symptomatic *P. vivax* infection, resulting in ameliorated disease severity compared to symptoms manifested during *P. falciparum* malaria. Whereas the aforementioned findings were observed in previously exposed symptomatic individuals in the field, malaria-naïve participants in controlled human malaria infection studies were found to mount stronger pro-inflammatory responses to *P. vivax* and elicit fever earlier compared to *P. falciparum* infection (Bach et al, 2023; Hemmer et al, 2006). Together, our findings suggest a rapid development of immunomodulatory mechanisms after a few exposures to *P. vivax* to reduce the impact of inflammatory responses responsible for clinical symptoms.

Our group previously identified an immunosuppressive transcriptional signature involved in the inhibition of CD4$^+$ T-cell function induced during asymptomatic *P. falciparum* malaria (Studniberg et al, 2022). In contrast, transcripts upregulated in the present study suggested successful T-cell differentiation and memory formation in *P. vivax* asymptomatic individuals. This may suggest that feedback mechanisms seemingly present to ameliorate excessive T-cell responses elicited during *P. vivax* infection appear to subside upon resolution of clinical symptoms. Whether these T-cell responses contribute to the control of infection requires further investigation. In the present study, the WGCNA tan module that was enriched for T-cell differentiation and TCR

rearrangement (both processes that were found to be upregulated by *P. vivax* asymptomatic carriers) positively correlated with populations of T$_{H2}$ and T$_{H17}$-polarized CD4$^+$ T cells, predicting reduced and increased risk of symptomatic infection, respectively (Ioannidis et al, 2021). These positive associations raise the possibility that while active, this T-cell response may not uniquely facilitate the elimination of circulating blood-stage parasites. Despite a transcriptional profile consistent with the successful development of T-cell responses, downregulation of transcriptional programs required for blood monocyte function was still apparent in asymptomatic *P. vivax* malaria, suggesting that this cellular compartment is altered even in response to low parasitemia infection.

Monocytes are instrumental in pathogen control and have been shown to not only inhibit *Plasmodium* replication (Dobbs et al, 2020), but a range of intracellular and extracellular bacteria (Lauvau et al, 2015), as well as viral infections (Sala and Kuka, 2020). A major finding of this study was a transcriptional signature reflective of profound monocyte perturbation in both symptomatic and asymptomatic *P. vivax* malaria. Whereas logistic regression analysis predicted dysregulation of critical transcriptional pathways controlling myeloid cell function, such as antigen presentation and TLR-associated signaling cascades during symptomatic *P. vivax* malaria, low parasitemia during asymptomatic infection appeared to be sufficient to compromise transcription of components required for formation and activation of the NLRP3 inflammasome. This reduction in monocyte transcriptional activity in response to *P. vivax* was aligned with a substantial depletion of a population of classical monocytes expressing various chemokine receptors, including CCR2 and CXCR4. Whereas CCR2 expression on monocytes has been associated with both mobilization from the bone marrow as well as recruitment to immune tissue (Tsou et al, 2007), expression of CXCR4 in monocytes has been associated with their localization in the bone marrow and identifies monocytes emerging from this tissue to replenish the mature monocyte pool (Chong et al, 2016). While it is possible that the depletion of CCR2$^+$CXCR4$^+$ classical monocytes in *P. vivax* malaria might reflect the migration of these cells to inflamed tissue, our findings do not exclude the intriguing possibility that *P. vivax* infection might have a detrimental effect in the differentiation and/or exit of monocytes from the bone marrow into peripheral blood, giving rise to a functionally impaired circulating monocyte pool. In support to that proposition, the bone marrow has been identified as an important parasite reservoir during *P. vivax* malaria (Obaldia et al, 2018), where the parasite seems to influence host erythropoiesis (Brito et al, 2022). Although further studies will be required to determine how the presence of *P. vivax* in the bone marrow affects monocyte homeostasis, our bioinformatic approach in the present study predicted downregulation of *CSF1*, which supports monocyte effector function and differentiation (MacDonald et al, 2010; Pierce et al, 1990), and the transcription factor *SPI1*—a positive regulator of monocyte differentiation (Rosa et al, 2007), as critical regulators of the dysregulated monocyte transcriptional signature observed in response to *P. vivax* malaria.

Unlike CCR2$^+$CXCR4$^+$ classical monocytes, non-classical monocytes were found to be abundant among symptomatic *P. vivax*-infected individuals. These cells play an integral role in recognizing and clearing pathogens as a first line of defense. It is accepted that opsonic phagocytosis is one of the main mechanisms contributing to protection against malaria

(Chiu et al, 2015; Hill et al, 2013), and non-classical monocytes have been found to be the most competent monocyte population for phagocytosing opsonized *P. falciparum* merozoites (Garcia-Senosiain, 2021) as well as *P. falciparum*-infected red blood cells (Vianou et al, 2024). It is therefore conceivable that these cells increase in response to high parasitemia *P. vivax* to reduce the burden of symptomatic infection.

Whole-genome co-expression analysis revealed critical associations between specific cell populations, transcriptional profiles, and their relationship with different infection outcomes, thereby allowing us to construct a comprehensive model of the immune response to *P. vivax* during natural infection. In this model, in addition to class-switched MBCs governing humoral immunity, inferred frequencies of classical monocytes clustering with gene modules enriched for myeloid cell function appeared to be associated only with healthy immune controls capable of controlling parasitemia, strongly suggesting that monocytes might play a critical role in the control of *P. vivax* parasite replication. Supporting this concept, it has been proposed that monocyte activation might be required to control clinical *P. vivax* infection, via production of pro-inflammatory cytokines such as IL-6, TNF-α, and IFN-γ (Kim et al, 2014), and phagocytosis of infected RBCs (Antonelli et al, 2014). The fact that these processes appeared to be compromised by the presence of *P. vivax* malaria raises concerns for the overall capacity of the host to fight infection, particularly in asymptomatic individuals that are usually unaware of their parasite carriage and therefore do not seek medical attention. Similar to our previous findings in asymptomatic *P. falciparum* malaria, our results here suggest that asymptomatic *P. vivax* infections are not innocuous and might lead to reduced innate responses to fight infection and/or respond to immunization. In an era of declining malaria transmission, further studies into the immune mechanisms underlying asymptomatic malaria infections of low parasite burden are necessary to inform policies for malaria control and elimination programs and vaccine deployment strategies. Herein, our findings have uncovered a dysregulated monocyte compartment as another deleterious consequence of asymptomatic malaria, suggesting that these infections are not benign, and providing a framework to consider screening and treatment of asymptomatic populations.

# Methods

### Reagents and tools table

| Reagent or resource | Source or reference | Identifier or catalog number |
|---|---|---|
| **Biological samples** | | |
| Melbourne unexposed healthy control | Walter and Eliza Hall Institute of Medical Research Volunteer Blood Donor Registry | N/A |
| Timika healthy immune control | Pigapu and Hirapau villages, Timika | HC |
| Timika asymptomatic *P. vivax* malaria | Pigapu and Hirapau villages, Timika | AM |
| Timika symptomatic *P. vivax* malaria | Rumah Sakit Mitra Masyarakat Hospital, Timika | SM |

| Reagent or resource | Source or reference | Identifier or catalog number |
|---|---|---|
| **Antibodies** | | |
| **ELISA antibodies** | | |
| HRP-conjugated streptavidin | Thermo Scientific | Cat.# N100 |
| Mouse anti-human TNF-alpha monoclonal | PeproTech | Cat.# 500-M26-500UG |
| Biotin-conjugated rabbit anti-human TNF-alpha polyclonal | PeproTech | Cat.# 500-P31ABT-50UG |
| Human TNF-alpha recombinant protein | PeproTech | Cat.# 300-01A-50UG |
| Mouse anti-human IL-1 beta monoclonal | PeproTech | Cat.# 500-M01B-500UG |
| Biotin-conjugated goat anti-human IL-1 beta polyclonal | PeproTech | Cat.# 500-P21BGBT-25UG |
| Human IL-1 beta recombinant protein | PeproTech | Cat.# 200-01B-10UG |
| **Surface marker antibodies for CyTOF** | | |
| 116 Cd-conjugated mouse anti-human CD45 | Fluidigm | Clone HI30, Cat.# 3116001B |
| 141 Pr-conjugated mouse anti-human CCR6 | Fluidigm | Clone G034E3, Cat.# 3141003A |
| 143 Nd-conjugated mouse anti-human CD45RA | Fluidigm | Clone HI100, Cat.# 3143006B |
| 144 Nd-conjugated mouse anti-human CCR5 | Fluidigm | Clone NP-6G4, Cat.# 3144007C |
| 147 Sm-conjugated mouse anti-human CD11c | Fluidigm | Clone 3.9, Cat.# 92J038147 |
| 150 Nd-conjugated mouse anti-human CD86 | Fluidigm | Clone IT2.2, Cat.# 3150020B |
| 152 Sm-conjugated mouse anti-human CD36 | Fluidigm | Clone 5-271, Cat.# 3152007B |
| 153 Eu-conjugated mouse anti-human CCR2 | Fluidigm | Clone K036C2, Cat.# 3153023B |
| 154 Sm-conjugated mouse anti-human CD163 | Fluidigm | Clone GHI/61, Cat.# 3154007B |
| 155 Gd-conjugated mouse anti-human CD56 | Fluidigm | Clone B159, Cat.# 3155008B |
| 156 Gd-conjugated mouse anti-human CXCR4 | Fluidigm | Clone 12G5, Cat.# 3156029B |
| 158 Gd-conjugated mouse anti-human CD10 | Fluidigm | Clone HI10a, Cat.# 3158011B |

| Reagent or resource | Source or reference | Identifier or catalog number |
|---|---|---|
| 159 Tb-conjugated mouse anti-human CCR7 | Fluidigm | Clone, G043H7, Cat.# 3159003A |
| 160 Gd-conjugated mouse anti-human CD14 | Fluidigm | Clone RMO52, Cat.# 3160006B |
| 161 Dy-conjugated mouse anti-human CD80 | Fluidigm | Clone B7-1, Cat.# 3161023B |
| 162 Dy-conjugated mouse anti-human APC | Fluidigm | Clone APC003, Cat.# 3162006B |
| 171 Yb-conjugated mouse anti-human CD9 | Fluidigm | Clone SN4 C3-3A2, Cat.# 3171009B |
| 172 Yb-conjugated mouse anti-human CX3CR1 | Fluidigm | Clone K0124E1, Cat.# 92J046172 |
| 173 Yb-conjugated mouse anti-human HLA-DR | Fluidigm | Clone L243, Cat.# 3173005B |
| 174 Yb-conjugated mouse anti-human CD4 | Fluidigm | Clone SK-3, Cat.# 3174004C |
| 209 Bi-conjugated mouse anti-human CD16 | Fluidigm | Clone 3G8, Cat.# 3209002B |
| APC-conjugated mouse anti-human CD3 | Biolegend | Clone SK7, Cat.# 981012 |
| APC-conjugated mouse anti-human CD19 | Biolegend | Clone HIB19, Cat.# 982406 |
| **Chemicals, enzymes, and other reagents** | | |
| Human TruStain FcX | Biolegend | Cat.# 422302 |
| Cell-ID Cisplatin | Fluidigm | Cat.# 201064 |
| Cell-ID Iridium Intercalator | Fluidigm | Cat.# 201192A |
| MaxPar water | Fluidigm | Cat.# 201069 |
| 4-Element EQ normalization beads | Fluidigm | Cat.# 201078 |
| BD Cytofix/ Cytoperm Fixation/ Permeabilization Kit | BD Biosciences | Cat.# 554714 |
| Phosphate-buffered saline | Gibco | Cat.# 20012050 |
| Fetal Bovine Serum | Gibco | Cat.# 10099141 |
| Bovine Serum Albumin | Sigma-Aldrich | Cat.# A7906-100G |
| RPMI 1640 Medium, GlutaMAX Supplement | Gibco | Cat.# 61870127 |
| Penicillin-streptomycin | Gibco | Cat.# 15140122 |
| Trypan Blue | Sigma-Aldrich | Cat.# T8154-100ML |

| Reagent or resource | Source or reference | Identifier or catalog number |
|---|---|---|
| **Software** | | |
| featureCounts | Liao et al, 2014 | https://www.bioconductor.org/packages/release/bioc/html/Rsubread.html |
| FlowJo v10.10 | BD Biosciences | https://www.flowjo.com/flowjo10/overview |
| glmnet | Friedman et al, 2010 | https://CRAN.R-project.org/package=glmnet |
| GraphPad Prism v10.0 | GraphPad | https://www.graphpad.com/ |
| Ingenuity Pathway Analysis | QIAGEN | https://digitalinsights.qiagen.com/products-overview/discovery-insights-portfolio/analysis-and-visualization/qiagen-ipa/ |
| limma | Ritchie et al, 2015 | http://bioconductor.org/packages/release/bioc/html/limma.html |
| pheatmap | NA | https://CRAN.R-project.org/package=pheatmap |
| psych | Revelle, 2021 | https://CRAN.R-project.org/package=psych |
| Rsubread | Liao et al, 2019 | http://bioconductor.org/packages/release/bioc/html/Rsubread.html |
| RUVSeq | Risso et al, 2014 | https://bioconductor.org/packages/release/bioc/html/RUVSeq.html |
| tmod | Zyla et al, 2019 | https://CRAN.R-project.org/package=tmod |
| Weighted Correlation Network Analysis | Langfelder and Horvath, 2008 | https://CRAN.R-project.org/package=WGCNA |
| **Published data** | | |
| Haemopedia RNA-seq data | Choi et al, 2019 | https://www.haemosphere.org/datasets/show |
| *P. falciparum* RNA-seq data | Studniberg et al, 2022 | GEO series (GSE181179) |
| *P. vivax* CyTOF data | Ioannidis et al, 2021 | PMID: 34128836 |
| *P. vivax* RNA-seq data | This paper | GEO series (GSE273483) |
| **Reagent kits and Instruments** | | |
| Agilent TapeStation 2200 | Agilent | N/A |
| Absorbance 96 plate reader | Byonoy | N/A |
| EasySep Human Monocyte Enrichment Kit without CD16 Depletion | STEMCELL Technologies | Cat.# 19058 |

| Reagent or resource | Source or reference | Identifier or catalog number |
|---|---|---|
| Helios mass cytometer | Fluidigm | N/A |
| Illumina NextSeq 500 | Illumina | N/A |
| Illumina TruSeq RNA Library Prep Kit (<100 ng) | Illumina | Cat# RS-122-2001 |
| RNeasy Plus Mini Kit | QIAGEN | Cat# 74134 |

## Study design

A cross-sectional study was conducted in the Timika region of Papua, Indonesia. Full details of the study have been previously described (Ioannidis et al, 2021). Briefly, participants (aged between 5-46 years) donated a 10 ml venous blood sample at enrollment, and PBMCS and plasma were frozen. Parasite densities and the species of infection were determined by light microscopy examination of Giemsa-stained blood smears. In addition, 29 presenting with malaria at the Rumah Sakit Mitra Masyarakat Hospital were enrolled in the study. Symptomatic malaria was defined as the presence of auxiliary fever of at least 37.5 °C, chills, malaise, headache, or vomiting at the time of examination or up to 24 h prior to examination and the presence of a *P. vivax*-positive blood smear and no other cause of fever discernible by physical exam. Symptomatic individuals ($n = 40$) included in the study had greater than 150 parasites/µL blood. Individuals with a *P. vivax*-positive blood smear and no clinical symptoms were classified as having asymptomatic infections ($n = 34$). Healthy immune controls ($n = 31$) had a negative light microscopy and PCR diagnosis. Previous exposure to malaria in these individuals was confirmed by ELISA against DBP recombinant protein (Ioannidis et al, 2021). The study was approved by the human research ethics committees of the Eijkman Institute for Molecular Biology, the Walter and Eliza Hall Institute of Medical Research, Monash University, and the Northern Territory Department of Health & Families and Menzies School of Health Research. Written informed consent was obtained from all participants. The experiments conformed to the principles set out in the WMA Declaration of Helsinki and the Department of Health and Human Services Belmont Report.

## RNA-sequencing

Samples with sufficient material to allow assessment of multiple endpoints were selected for RNA was extracted from $2 \times 10^5$ PBMCs from with the RNeasy Plus Mini Kit (QIAGEN Inc.) following the manufacturer's instructions and quantified using RNA ScreenTape on the Agilent TapeStation 2200 System. Libraries were prepared with either 50 ng or 25 ng of total RNA using the Illumina TruSeq RNA Library Prep Kit (<100 ng) v1.0 following the manufacturer's instructions. Products were checked for size using D1000 ScreenTape on the Agilent TapeStation 2200 system, pooled in equimolar amounts, and submitted for sequencing by paired end, 80 base pair reads on an Illumina NextSeq 500 platform.

## Differential expression analysis

Raw sequence reads in FASTQ file format were aligned to the human reference genome GRC38/hg38 (GRCh38.p13) using the align function in Rsubread (Liao et al, 2019) v2.10.5 with default parameters. Fragments of aligned sequences overlapping human genes were quantified using featureCounts (Liao et al, 2014). Genes were identified using Gencode v42 primary assembly annotation. Genes with no symbols, duplicated genes, and sex-linked genes (*XIST* gene and genes uniquely located on chromosome Y) were filtered out of the analysis. Hemoglobin genes were found to be highly variable and were also filtered out. Expression-based filtering was performed using the filterByExpr function (Chen et al, 2016) in edgeR (Robinson et al, 2010) with default parameters. Therefore, genes with a minimum of 10 counts-per-million reads in at least 9 samples (the minimum number of samples per group in this experiment) were retained, leaving 15,780 genes for differential expression analysis. Sample composition was normalized using the trimmed mean of M-values method (Robinson and Oshlack, 2010) in edgeR. Known and unknown variation in the data was removed using the remove unwanted variation using residuals (RUVr) function in RUVSeq (Risso et al, 2014) v1.34.0 setting $k = 5$. This analysis was performed using the 1000 least-significant genes from a fist-pass generalized linear model regression on the normalized counts as negative control genes. The W produced from this analysis was extracted and included in linear models. Counts were then transformed to log2-counts per million using the voom function (Law et al, 2014) in limma v3.56.2 (Ritchie et al, 2015) with associated precision weights. Linear models incorporating the W matrix extracted from RUVr were fitted to each gene, and differential expression assessed using robust empirical Bayes moderated *t*-statistics (Smyth, 2004). The FDR was controlled to below 1% using the method of Benjamini and Hochberg. Cell-type deconvolution was performed on transcripts per million of the filtered and normalized counts using the CIBERSORTx tool (Newman et al, 2019) with B-mode batch correction, maximum permutations, and PMBCs from the LM22 dataset (Newman et al, 2015) as the signature matrix file. Cell-type marker genes were identified, and monocyte frequencies were inferred using the dtangle package (Hunt et al, 2019) with the Haemopedia RNA-seq dataset (Choi et al, 2019) as a reference.

## Pathway analysis

Enrichment of DE genes between pairwise comparisons in blood transcriptional modules was determined using tmodLimmaTest() in the tmod package v0.50.13 (Zyla et al, 2019) with an FDR of 1% and otherwise default parameters. Up and downregulated genes for each module were determined using the tmodLimmaDecideTests function in tmod with a *P* value of 0.01 and no log-fold-change threshold. Pathway analysis was conducted to determine over-representation of DE genes from each pairwise comparison in GO terms and KEGG pathways using the goana and kegga functions (Young et al, 2010) in limma. Ingenuity Pathway Analysis software (QIAGEN Inc.) was used for canonical pathway, upstream regulator, and disease and function analyses.

## Network analysis

Gene co-expression networks were constructed with the WGCNA (Langfelder and Horvath, 2008) package v1.72-1 using the filtered,

normalized, and transformed gene expression data. Weighted gene network adjacencies were calculated using a signed network to identify genes with positively correlated expression patterns. Co-expression similarity was raised to a soft-thresholding power of 18, which was identified as the most appropriate soft-thresholding power for a signed network using the pickSoftThreshold function. Gene co-expression adjacencies were converted to a topological overlap matrix (TOM), and corresponding dissimilarities were calculated. Hierarchical clustering of the TOM produced a dendrogram, whereby interconnected and densely grouped branches represented genes with high positive co-expression similarity. Gene co-expression modules were identified using the recommended minimum module size of 30. A dendrogram cut height of 0.35 was applied to merge modules containing genes with similar co-expression profiles, resulting in a total of 15 gene co-expression modules. Gene co-expression modules with significant enrichment in the DE analysis were identified by testing gene modules against each pairwise contrast in the *P. vivax* DE experiment with the mroast function (Wu et al, 2010) in limma, with a maximum number of rotations for robust *P* value calculation. Correlations between module eigengenes, module hub genes, clinical traits, cell populations associated with increased or reduced odds of *P. vivax* malaria (Ioannidis et al, 2021) and clinical groups were calculated using the Spearman's rank correlation coefficient in the psych package v2.1.9. Correlations with a Spearman correlation coefficient above 0.35 were visualized using MetScape v3.1.3 (Basu et al, 2017) in Cytoscape v3.10.0 (Shannon et al, 2003).

## CyTOF

PBMCs (up to $2 \times 10^6$) from *P. vivax* symptomatic and asymptomatic infected individuals, as well as healthy immune controls, were stained with 5 µM Cell-ID Cisplatin (Fluidigm) in phosphate-buffered saline (PBS; Gibco) for 5 min at room temperature (RT). Cells were washed with CyTOF staining buffer (PBS with 0.5% bovine serum albumin [BSA; Sigma]) and blocked with Human TruStain FcX (Biolegend). Cells were stained with a primary cocktail consisting of APC-conjugated CD3 (clone Sk7; Biolegend) and APC-conjugated CD19 (clone HIB19; Biolegend) in CyTOF staining buffer for 30 min at RT, washed, and then stained with a secondary cocktail of metal conjugated surface marker antibodies (Reagents and Tools Table) in CyTOF staining buffer for 30 min at RT on a shaker. After surface staining, cells were fixed and permeabilized with BD Cytofix/Cytoperm Fixation/Permeabilization Kit (BD). Cells were then washed twice and stored in Cytofix/Cytoperm buffer (BD) with 125 nM Cell-ID iridium intercalator (Fluidigm) for 2 weeks at 4 °C. For long-term storage the samples were then resuspended in 90% FBS (Gibco) 10% Dimethyl sulfoxide (DMSO; Sigma) and frozen at −80 °C. Prior to data acquisition, cells were thawed slowly at 4 °C, washed once with ultrapure water (1 ml), once with CAS (1 ml) once with ultrapure water (2 ml), then resuspended in a 1/10 dilution of 4-Element EQ normalization beads (Fluidigm) in ultrapure water. The samples were then passed through a cell strainer (Corning CAT# 352235). Cells were analyzed on a Helios model mass cytometer (Fluidigm) at ~300 events per second. Data were normalized using the signal from 4-Element EQ Beads (Fluidigm) as previously described (Finck et al, 2013). Manual gating was performed using FlowJo version 10.10

(BD Biosciences) to exclude doublets and dead cells before monocyte cell populations were selected and exported for further analysis in Cytobank (Kotecha et al, 2010). Monocyte populations were then visualized using opt-SNE (Belkina et al, 2019), while FlowSOM (Gassen et al, 2015) was used to identify monocyte subpopulations. The following parameters were included in the opt-SNE analysis CD16, CCR2, CXCR4, CD10, CD45RA, CCR5, CD86, CD11c, CD36, CD163, CD9, CX3CR1, HLA-DR, CD4.

## Monocyte isolation and stimulation

Monocytes were isolated using the EasySep Human Monocyte Enrichment Kit without CD16 Depletion (STEMCELL Technologies) as per the manufacturer's instructions. The isolated monocytes were resuspended in RPMI media with GlutaMAX (Gibco) supplemented with 10% FBS (Gibco), 100 U/mL penicillin/Streptomycin (Gibco) and HEPES (Gibco). The isolated monocytes (200 µl of $1 \times 10^5$ cells/mL) were plated, with or without 50 ng/mL of LPS, in 96-well plates. The cells were then incubated at 37 °C with 5% $CO_2$ for 24 h, after which supernatants were collected and stored at −20 °C.

## ELISA

The concentration of TNF-α and IL-1β in cell culture supernatants was determined by ELISA. Briefly, maxisorb 96-well plates (Nunc, cat#442404) were coated with 2 µg/mL of either TNF-α and IL-1β capture antibody (PeproTech) overnight at 4 °C. The plate was washed with PBST (PBS with 0.05% Tween 20) and blocked with 5% BSA/PBST for 2 h at room temperature. The plate was washed, samples and either TNF-α or IL-1β standards (PeproTech) were added and incubated for 2 h at room temperature. The plate was washed, and 1 µg/mL of biotin-conjugated secondary antibody (PeproTech) was added and incubated for 1.5 h at room temperature. The plate was washed, and 0.25 µg/mL of HRP-Conjugated Streptavidin (Thermo Scientific) was added and incubated for 45 min at room temperature. The plate was washed, and KPL TMB substrate (Sera Care) added and incubated in the dark and stopped by acidification. The optical density was determined at 450 nm on the Byonoy 96-well plate reader. A four-parameter logistic (4PL) curve was performed and the concentrations interpolated using GraphPad Prism v10.

## Statistical analysis

Characteristics of clinical groups were compared using ordinary one-way ANOVA with Tukey's multiple comparisons for continuous data that were normally distributed, while the Kruskal–Wallis test was used for data that did not follow normal distribution. Gene-set enrichment of *P. falciparum* DE genes (Studniberg et al, 2022) against the *P. vivax* dataset was tested using the roast function in limma with a maximum number of rotations. Expression data for heatmaps were scaled using the scale_rows function in pheatmap v1.0.12, and heatmaps were generated using default clustering parameters in ComplexHeatmap (Gu et al, 2016) v2.16.0. A generalized logistic regression model was fitted using elastic net regularized regression following the glmnet (v4.1-8) pipeline in R (Friedman et al, 2010). The hyperparameter α was tuned to 0.9 by applying leave-one-out cross-validation (LOOCV)

over a range of $\alpha$ at increments of 0.05, allowing for a robust elastic net regularization approach by combining both L1 and L2 penalties. Prevalidated predictions were retained for selecting the minimum $\lambda$ value. Predictor classification was calculated using glmnet and visualized using ggplot2 (Wickham, 2016) v3.5.0 (Montojo et al, 2010) and visualized using a prefuse force-directed layout in Cytoscape (Shannon et al, 2003). The FDR was controlled to below either 1% or 5% as indicated using the method of Benjamini and Hochberg. Statistical analyses were performed in GraphPad Prism v9 and R v4.3.1. No blinding was performed in this study.

## Data availability

The datasets and computer code produced in this study are available in the following databases: Gene-level read counts (raw counts): Gene Expression Omnibus GSE273483. Original code: GitHub (https://github.com/dhansenlab/pvivax-bulk-rnaseq-timika).

The source data of this paper are collected in the following database record: biostudies:S-SCDT-10_1038-S44320-025-00135-z.

## Peer review information

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

## Acknowledgements

This work was performed in part at Sydney Cytometry, a platform established through collaboration between the University of Sydney and the Centenary Institute. Supported by the Australian Government National Health and Medical Research Council IRIISS, Ideas Grant 2028898 (DSH), e-Asia Joint Research Program Grant 2026175 (DSH), and Program Grant GNT1132975 (RNP); the Australian Academy of Science Regional Collaboration Grant (DSH), the Victorian State Government Operational Infrastructure Support (DSH) and LifeArc, UK (DSH).

## Author contributions

**Stephanie I Studniberg**: Conceptualization; Data curation; Formal analysis; Validation; Investigation; Visualization; Methodology; Writing—original draft; Writing—review and editing. **Mariam Bafit**: Investigation; Methodology. **Lisa J Ioannidis**: Formal analysis; Investigation. **Matthew J Worley**: Investigation. **Leily Trianty**: Resources. **Retno A S Utami**: Resources. **Agatha M Puspitasari**: Resources. **Dwi Apriyanti**: Resources. **Farah N Coutrier**: Resources. **Jeanne R Poespoprodjo**: Resources. **Enny Kenangalem**: Resources. **Benediktus Andries**: Resources. **Pak Prayoga**: Resources. **Ric N Price**: Resources. **Rintis Noviyanti**: Conceptualization; Resources; Supervision; Investigation; Methodology; Project administration. **Alexandra L Garnham**: Formal analysis; Supervision. **Diana S Hansen**: Conceptualization; Data curation; Formal analysis; Supervision; Funding acquisition; Investigation; Methodology; Writing—original draft; Project administration; Writing—review and editing.

Source data underlying figure panels in this paper may have individual authorship assigned. Where available, figure panel/source data authorship is listed in the following database record: biostudies:S-SCDT-10_1038-S44320-025-00135-z.

## Disclosure and competing interests statement

The authors declare no competing interests.

# Expanded View Figures

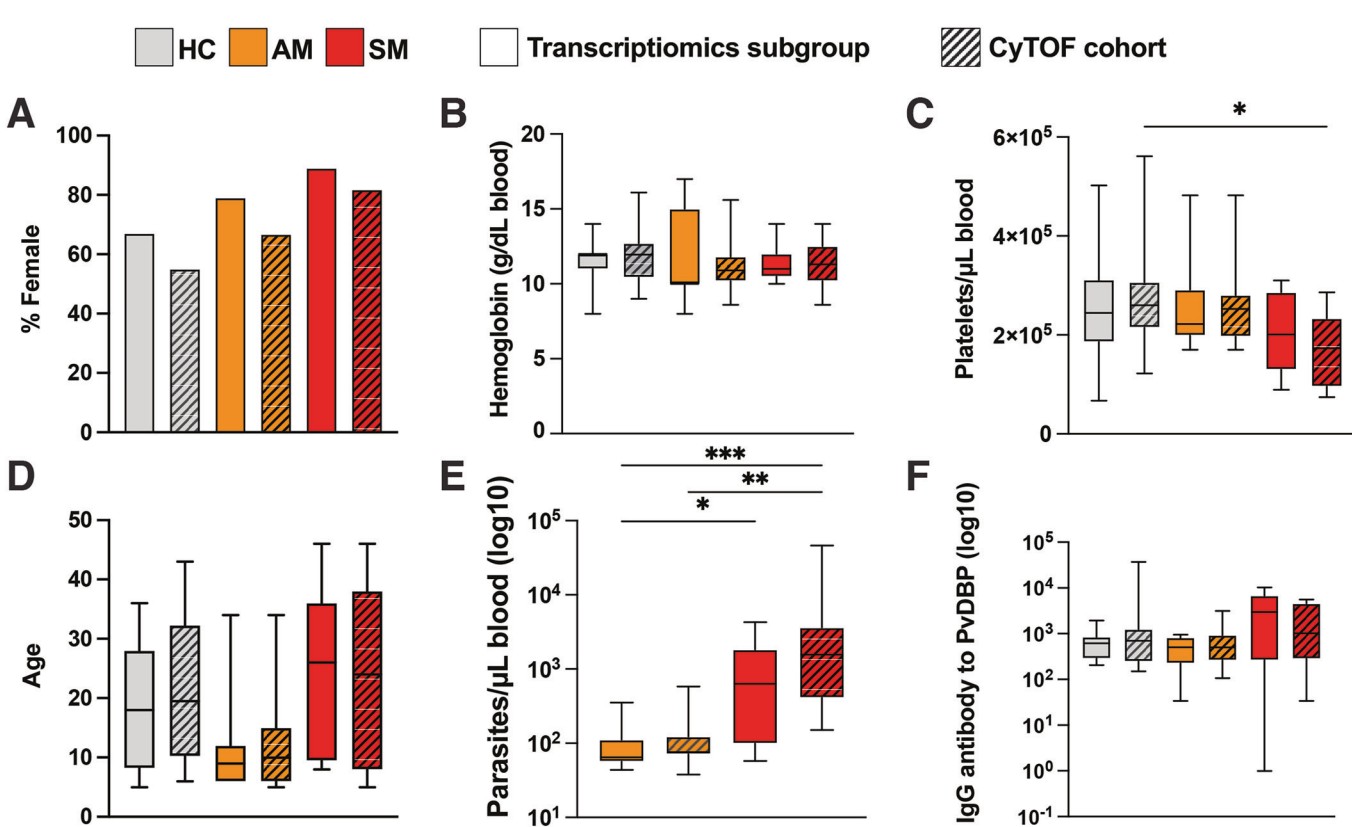

**Figure EV1. CyTOF cohort and transcriptomics sub-cohort characteristics.**

(A–F) Immune phenotyping by CyTOF was conducted with *P. vivax* symptomatic ($n = 11$ SM) and *P. vivax* asymptomatic ($n = 19$ AM) infected participants as well as *P. vivax* negative healthy community controls ($n = 24$ HC) (striped bars). Transcriptional profiling was conducted with a sub-cohort of *P. vivax* symptomatic ($n = 9$) and *P. vivax* asymptomatic ($n = 11$) infected participants, as well as *P. vivax* negative healthy immune control ($n = 12$) (full bars). Clinical parameters determined in the study include gender (A), hemoglobin (g/dL blood) (B), platelets/µL blood (*$P = 0.0372$) (C), age (D), parasite density (*$P = 0.0177$, **$P = 0.0011$, ***$P = 0.0002$) (E), and IgG antibody to *P. vivax* Duffy binding protein (F). Boxes represent the 25th to 75th percentile, whiskers show the range (minimum to maximum), and lines represent the median. Significance was determined by the Chi-square test (A) and the Kruskal–Wallis test with Dunn's multiple comparisons (B–F).

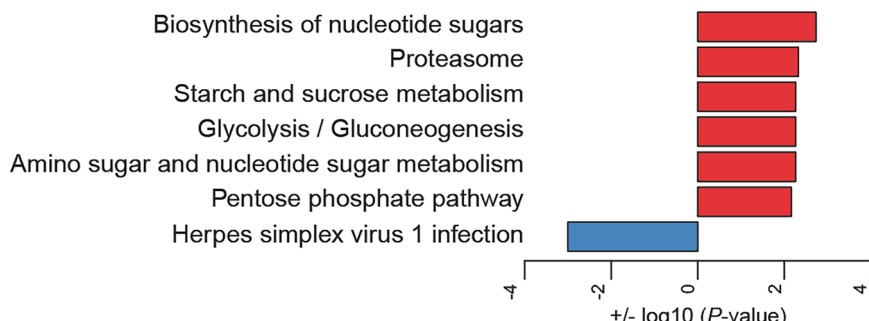

**Figure EV2. Kyoto Encyclopedia of Genes and Genomes (KEGG) pathway analysis was conducted for genes DE between symptomatic and asymptomatic *P. vivax* malaria.**

Gene expression profiles of PBMCs from symptomatic (n = 9) and asymptomatic (n = 11) *P. vivax*-infected individuals were compared. Bar plots showing significantly enriched KEGG pathways identified using hypergeometric testing implemented with the kegga function in limma and scaled by + or − log10 (P value). Red KEGG pathways are enriched in symptomatic malaria and blue KEGG pathways are enriched in asymptomatic malaria.

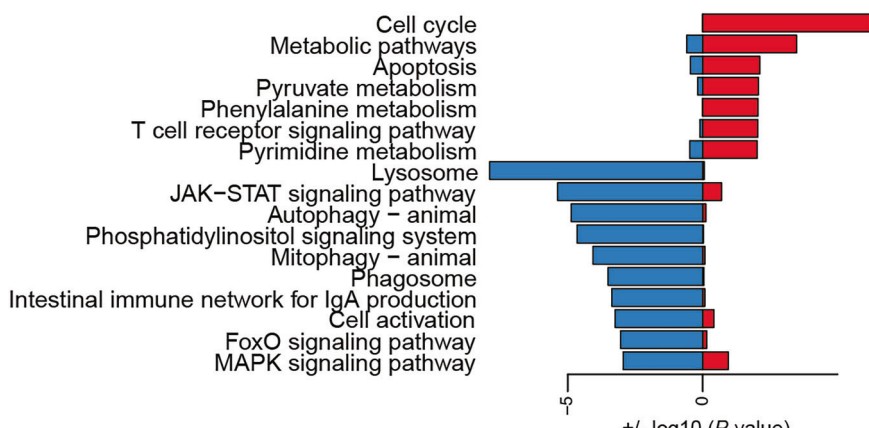

**Figure EV3. Kyoto Encyclopedia of Genes and Genomes (KEGG) pathway analysis was conducted for genes DE between symptomatic *P. vivax* malaria and healthy community controls.**

Gene expression profiles of PBMCs from symptomatic ($n = 9$) *P. vivax*-infected individuals and healthy immune controls ($n = 12$) were compared. Bar plots showing significantly enriched KEGG pathways identified using hypergeometric testing implemented with the kegga function in limma and scaled by $+$ or $-$log10 ($P$ value). Red KEGG pathways are upregulated and blue KEGG pathways are downregulated in symptomatic malaria.

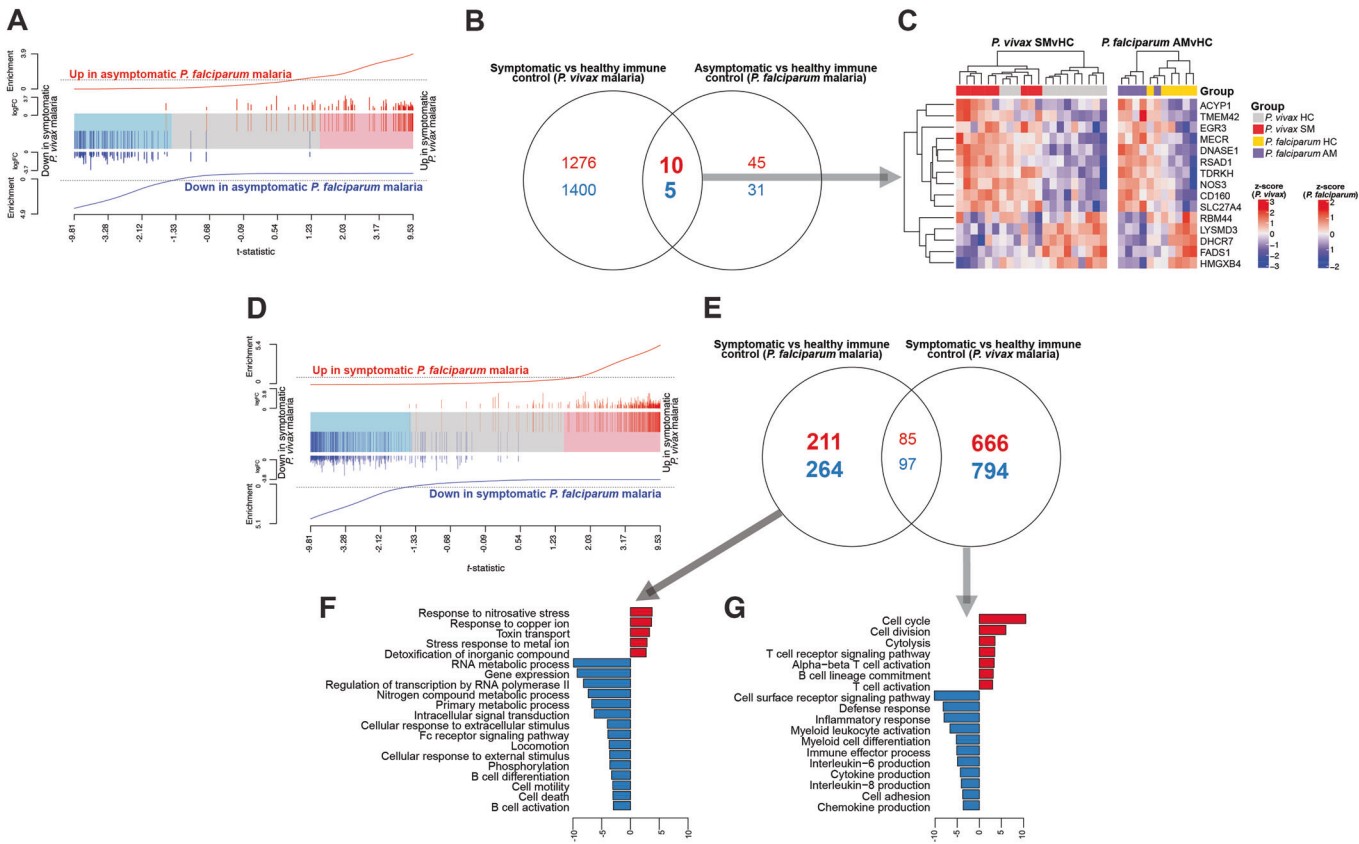

**Figure EV4. Symptomatic *P. vivax* malaria induces immunoregulatory transcripts and a reduced inflammatory response than symptomatic *P. falciparum* malaria.**

Gene expression profiles between symptomatic *P. vivax* malaria and healthy immune controls were compared with asymptomatic and symptomatic *P. falciparum* malaria and healthy immune controls (Studniberg et al, 2022). (A) Barcode plot showing significant positive correlation ($P < 0.0001$) between genes regulated in asymptomatic *P. falciparum* malaria vs healthy immune controls against genes regulated in symptomatic *P. vivax* malaria vs healthy immune controls. Significance was tested using the roast test in limma. (B) Venn diagram showing the overlap of differentially expressed genes between symptomatic *P. vivax* vs healthy immune controls, and asymptomatic *P. falciparum* vs healthy immune controls. A total of 15 genes were co-regulated by both symptomatic *P. vivax* and asymptomatic *P. falciparum* malaria. (C) Heatmap showing expression of the 15 genes co-regulated between symptomatic *P. vivax* and asymptomatic *P. falciparum* malaria identified in (B) against healthy immune controls in the *P. vivax* study on the left, and the *P. falciparum* study on the right. (D) Barcode plot showing significant positive correlation ($P < 0.0001$) between genes regulated in symptomatic *P. falciparum* malaria vs healthy immune controls against genes regulated in symptomatic *P. vivax* malaria vs healthy immune controls. Significance was tested using the roast test in limma. (E) Venn diagram showing the overlap of differentially expressed genes between symptomatic *P. vivax* vs healthy immune controls, and symptomatic *P. falciparum* vs healthy immune controls. A total of 1460 genes were uniquely regulated by symptomatic *P. vivax* malaria, and a total of 475 genes were uniquely regulated by symptomatic *P. falciparum* malaria. (F, G) Bar plots showing significantly enriched gene ontology (GO) terms scaled by $+/-$ log10 (P value) in genes uniquely regulated by symptomatic *P. vivax* malaria (F) and genes uniquely regulated by symptomatic *P. falciparum* malaria (G) identified using hypergeometric testing implemented with the goana function in limma and scaled by $+$ or $-$log10 (P value). Red GO terms are upregulated, and blue GO terms are downregulated in symptomatic malaria against healthy immune controls.

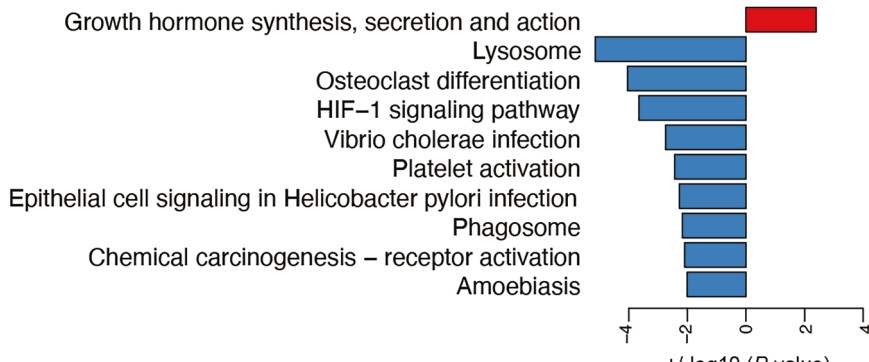

**Figure EV5. Kyoto Encyclopedia of Genes and Genomes (KEGG) pathway analysis was conducted for genes DE between asymptomatic *P. vivax* malaria and healthy community controls.**

Gene expression profiles of PBMCs from asymptomatic ($n = 11$) *P. vivax*-infected individuals and healthy immune controls ($n = 12$) were compared. Bar plots showing significantly enriched KEGG pathways identified using hypergeometric testing implemented with the kegga function in limma and scaled by + or −log10 (P value). Red KEGG pathways are upregulated and blue KEGG pathways are downregulated in asymptomatic malaria.

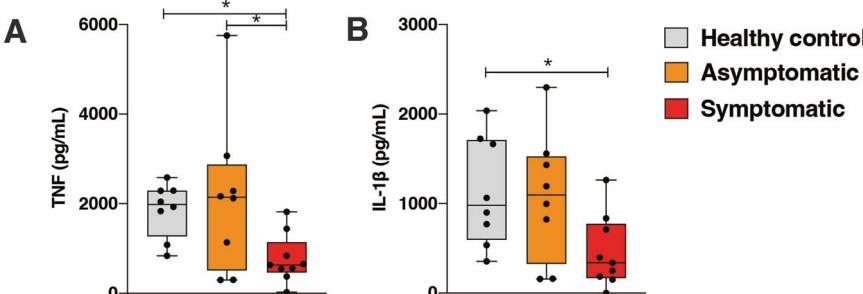

**Figure EV6. Reduced cytokine responses in monocytes from *P. vivax*-infected individuals relative to healthy immune controls.**

Blood monocytes were isolated from PBMCs of *P. vivax* symptomatic ($n = 8$) and *P. vivax* asymptomatic ($n = 8$) infected participants, as well as *P. vivax* negative healthy immune controls ($n = 8$). Cells were then plated at a density of $1 \times 10^5$ cell/ml and stimulated with LPS (50 ng/ml) for 24 h. (**A, B**) Cell culture supernatants were harvested and their TNF (**A**) and IL-1β (**B**) levels were measured by capture ELISA. Boxes represent the 25th to 75th percentile, whiskers show the range (minimum to maximum), and lines represent the median. Significance was determined by Kruskal–Wallis test with Dunn's multiple testing. *$P = 0.0101$ for TNF values between symptomatic individuals and healthy controls, *$P = 0.0288$ for TNF values between symptomatic individuals and asymptomatic individuals, *$P = 0.0226$ for IL-1β values between symptomatic individuals and healthy controls.

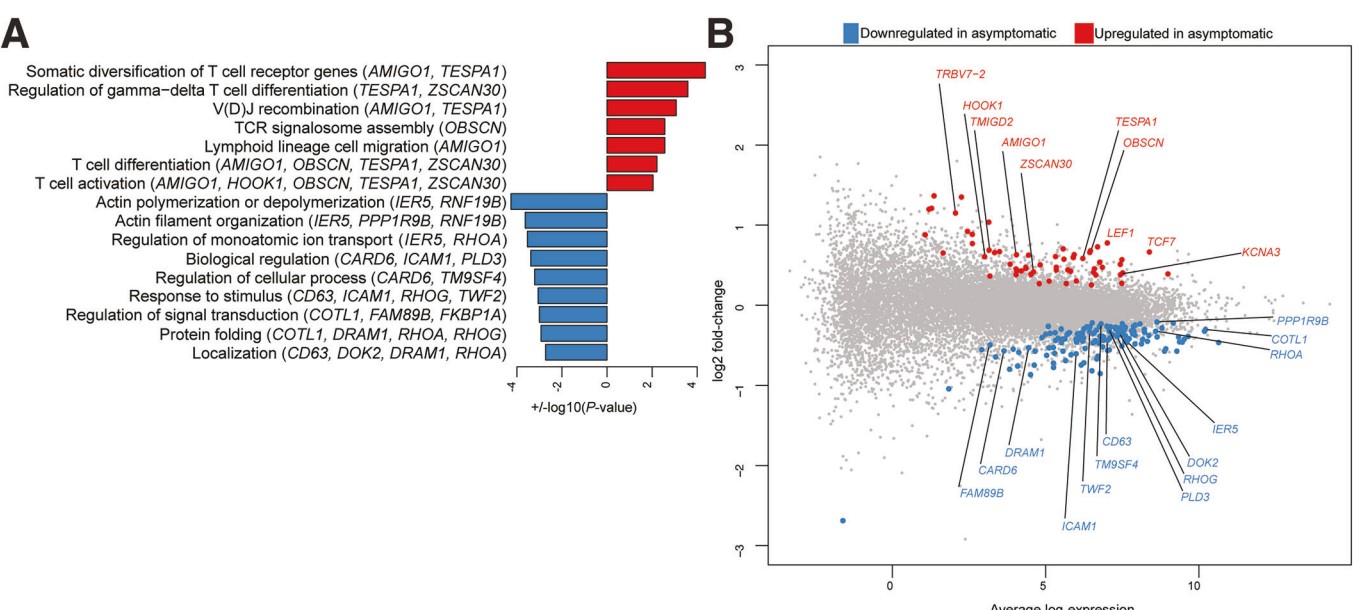

**Figure EV7. *P. vivax* asymptomatic malaria is permissive of normal T-cell differentiation but compromises immune effector function.**

Gene expression profiles for 83 genes uniquely regulated by *P. vivax* asymptomatic (AM) infected participants and healthy immune controls (HC). (A) Bar plots showing significantly enriched gene ontology (GO) terms identified using hypergeometric testing implemented with the goana function in limma and scaled by + or − log10 (P value). Red GO terms are upregulated, and blue GO terms are downregulated in AM. (B) Mean-difference plot displaying DEGs between AMvHC. Each gene is plotted as a single point determined by Log2-fold change and average transcript abundance. Red genes are overrepresented, and blue genes are underrepresented in AM.

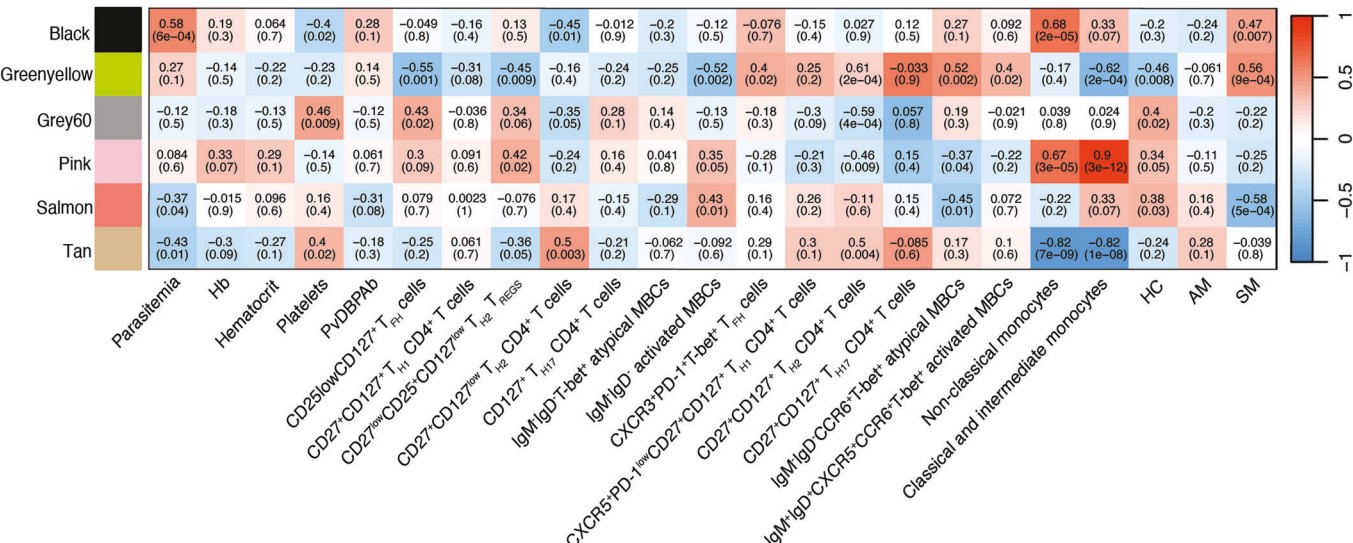

**Figure EV8. Weighted gene co-expression network analysis of the immune response to *P. vivax* malaria.**

Co-expression network analysis with whole-genome expression profiles of peripheral blood mononuclear cells from *P. vivax* symptomatic (SM, n = 9), asymptomatic (AM, n = 11) and healthy control (HC, n = 12) individuals. Expanded correlation heatmap from Fig. 6B, depicting associations between gene co-expression modules significantly enriched for at least one pairwise contrast in the limma-voom differential expression analysis, clinical traits, cell populations associated with increased or reduced odds of *P. vivax* malaria (Ioannidis et al, 2021), monocyte frequencies estimated by cell-type deconvolution and clinical groups. Red represents positive correlations and blue represents negative correlations.

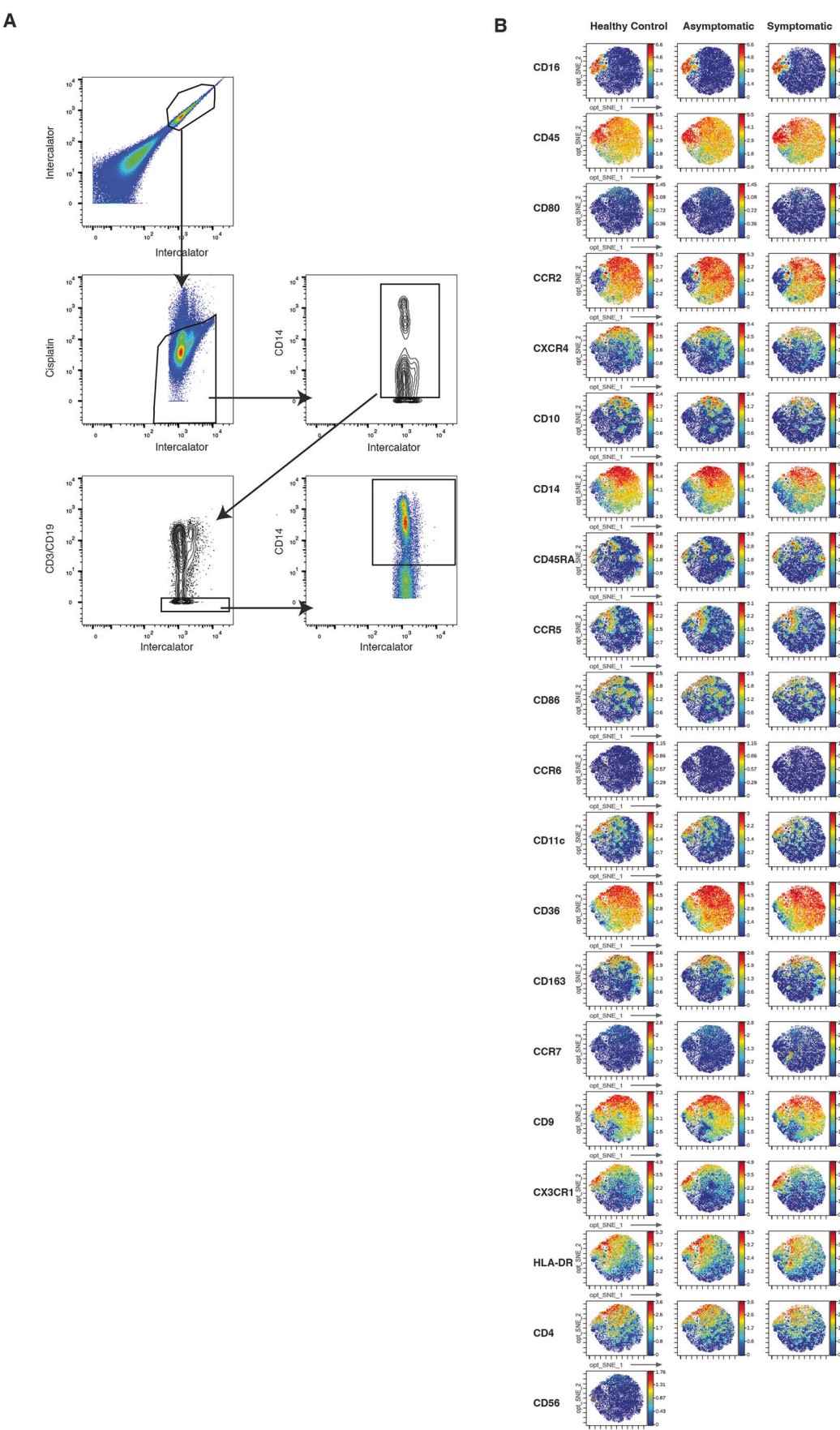

**Figure EV9.  Gating strategy for analysis of blood monocytes by CyTOF.**

Peripheral blood mononuclear cells from *P. vivax* symptomatic ($n = 8$), asymptomatic ($n = 8$), and healthy controls ($n = 8$) were stained with a panel of metal-labeled antibodies and analyzed CyTOF. (**A**) Manual gating was used to select for CD14$^+$ monocytes. (**B**) vi-SNE analysis was performed on pooled monocytes for each group and the cell surface expression was illustrated by their density.

