## [Peer Review File · Molecular Systems Biology]

Systems approach identifies monocyte imbalance in symptomatic and asymptomatic *P. vivax* malaria

Stephanie Studniberg, Mariam Bafit, Lisa Ioannidis, Matthew Worley, Leily Trianty, Retno Utami, Agatha Puspitasari, Dwi Apriyanti, Farah Coutrier, Jean Poespoprodjo, Enny Kenangalem, Benediktus Andries, Pak Prayoga, Ric Price, Rintis Noviyanti, Alexandra Garnham, and Diana Hansen

Corresponding author(s): Diana Hansen (diana.hansen@monash.edu)

Review Timeline:

Submission Date:	4th Sep 24
Editorial Decision:	11th Oct 24
Revision Received:	20th May 25
Editorial Decision:	1st Jul 25
Revision Received:	18th Jul 25
Accepted:	22nd Jul 25

Editor: Poonam Bheda

Transaction Report:

11th Oct 2024

Manuscript Number: MSB-2024-12610

Title: Systems approach identifies monocyte dysfunction in symptomatic and clinically silent *P. vivax* malaria

Dear Prof Hansen,

Thank you for the submission of your manuscript to Molecular Systems Biology. We have now received feedback from the three reviewers who agreed to evaluate your manuscript. As you will see from the reports below, the referees acknowledge the interest of the study and are overall supportive of your work; however they also comment on multiple aspects of the manuscript that should be strengthened in a revision. In particular, both reviewers 2 and 3 comment that there is a lack of experimental validation of the bioinformatic analyses in the manuscript. This aspect of the reviewer reports reflect our own initial editorial impression that the analysis could be an interesting contribution to the field, but that without any experimental validation supporting the conclusions the manuscript was somewhat preliminary. Therefore this would be an important point to address during the revision of your manuscript. All other issues raised would need to be satisfactorily addressed. Please let me know in case you would like to discuss in further detail any of the comments, I would be happy to schedule a call.

We require:

1) A .docx formatted version of the manuscript text (including legends for main figures, EV figures and tables). Please make sure that the changes are highlighted to be clearly visible. Alternatively you may choose to submit your manuscript as a LaTeX file.

4) A .docx formatted letter INCLUDING the reviewers' reports and your detailed point-by-point responses to their comments. As part of the EMBO Press transparent editorial process, the point-by-point response is part of the Peer Review File (PRF), which will be published alongside your paper.

5) A complete author checklist, which you can download from our author guidelines (<https://www.embopress.org/page/journal/17574684/authorguide#submissionofrevisions>). Please insert information in the checklist that is also reflected in the manuscript. The completed author checklist will also be part of the PRF.

6) Please note that all corresponding authors are required to supply an ORCID ID for their name upon submission of a revised manuscript.

7) It is mandatory to include a 'Data Availability' section after the Materials and Methods. Before submitting your revision, primary datasets produced in this study need to be deposited in an appropriate public database, and the accession numbers and database listed under 'Data Availability'. Please remember to provide a reviewer password if the datasets are not yet public (see <https://www.embopress.org/page/journal/17574684/authorguide#dataavailability>).

This study includes no data deposited in external repositories.

8) All Materials and Methods need to be described in the main text using our 'Structured Methods' format, which is required for all research articles. According to this format, the Methods section includes a Reagents and Tools Table (listing key reagents, experimental models, software and relevant equipment and including their sources and relevant identifiers) followed by a Methods and Protocols section describing the methods using a step-by-step protocol format. The aim is to facilitate adoption of the methodologies across labs. Please upload the Reagents and Tools table as a separate document when submitting your revised manuscript. More information on how to adhere to this format as well as a downloadable template (.docx) for the Reagents and Tools Table can be found in our author guidelines:

<https://www.embopress.org/page/journal/17444292/authorguide#structuredmethods>

An example of a Method paper with Structured Methods can be found here:
<https://www.embopress.org/doi/10.15252/msb.20178071>.

9) For data quantification: please specify the name of the statistical test used to generate error bars and P values, the number (n) of independent experiments (specify technical or biological replicates) underlying each data point and the test used to calculate p-values in each figure legend. The figure legends should contain a basic description of n, P and the test applied. Graphs must include a description of the bars and the error bars (s.d., s.e.m.). Please provide exact p values.

10) Our journal encourages inclusion of *data citations in the reference list* to directly cite datasets that were re-used and obtained from public databases. Data citations in the article text are distinct from normal bibliographical citations and should directly link to the database records from which the data can be accessed. In the main text, data citations are formatted as follows: "Data ref: Smith et al, 2001" or "Data ref: NCBI Sequence Read Archive PRJNA342805, 2017". In the Reference list, data citations must be labeled with "[DATASET]". A data reference must provide the database name, accession number/identifiers and a resolvable link to the landing page from which the data can be accessed at the end of the reference. Further instructions are available at .

11) We replaced Supplementary Information with Expanded View (EV) Figures and Tables that are collapsible/expandable online. A maximum of 5 EV Figures can be typeset. EV Figures should be cited as 'Figure EV1, Figure EV2' etc... in the text and their respective legends should be included in the main text after the legends of regular figures.

<https://www.embopress.org/page/journal/17574684/authorguide#expandedview>

13) Author contributions: CRediT has replaced the traditional author contributions section because it offers a systematic machine readable author contributions format that allows for more effective research assessment. Please remove the Authors Contributions from the manuscript and use the free text boxes beneath each contributing author's name in our system to add specific details on the author's contribution. More information is available in our guide to authors.

14) Disclosure statement and competing interests: We updated our journal's competing interests policy in January 2022 and request authors to consider both actual and perceived competing interests. Please review the policy

<https://www.embopress.org/competing-interests> and update your competing interests if necessary.

Please also suggest a striking image or visual abstract to illustrate your article as a PNG file 550 px wide x 300-600 px high. Share synopsis text and image, as well as eTOC:

Please note that these would be the final versions and changes during proofing are usually not allowed

16) As part of the EMBO Publications transparent editorial process initiative (see our policy here:

https://www.embopress.org/transparent-process#Review_Process), Molecular Systems Biology will publish online a Peer Review File (PRF) to accompany accepted manuscripts.

In the event of acceptance, this file will be published in conjunction with your paper and will include the anonymous referee reports, your point-by-point response and all pertinent correspondence relating to the manuscript. Let us know whether you agree with the publication of the PRF and as here, if you want to remove or not any figures from it prior to publication.

Please note that the Authors checklist will be published at the end of the PRF.

Molecular Systems Biology has a "scooping protection" policy, whereby similar findings that are published by others during review or revision are not a criterion for rejection. Should you decide to submit a revised version, I do ask that you get in touch after three months if you have not completed it, to update us on the status.

I look forward to receiving your revised manuscript.

Yours sincerely,

Poonam Bheda, PhD
Scientific Editor
Molecular Systems Biology

Reviewer #1:

Studniberg et al present a thorough analysis of transcriptional profiles of PBMCs collected from symptomatic and asymptomatic individuals infected with *P. vivax* alongside those of healthy controls from Papua in Indonesia.

The data is well explained and will be of interest to parasite immunologists, I suggest however a few points where I believe a more clearly could be afforded to the reader.

I believe the abstract misleads the reader to think that mass cytometry was done in the context of this study, and it should be clarified that the data used has already been published.

also in the abstract there is an unclear sentence "Despite allowing transcriptional profiles supporting T cell differentiation, a TH2 cell bias and class-switched memory B cells reducing risk of symptomatic infection, monocyte dysfunction persisted in asymptomatic individuals." I suggest rewording.

Throughout the manuscript several several sentences are very long and hard to follow. Between page 2 and 3 is a good example that could be easily cut into two to help the reader.

The first sentence of page 3 requires a reference, and should specifically mention if referring to Pv or Pf malaria.

The last sentence of the introduction needs a reference to tell the reader which data are the findings consistent with.

In the beginning of the results there is an unnecessary the repetition of the reference of the cohort. Saying once that it was already reported will suffice.

I suggest indicating the number of study participants analysed in the text of the result section, as well as the age range.

Fig 1F should have represented a line of something to indicate the negative control IgG level. How was it assessed that all individuals in the healthy control group exhibited previous exposure to the *P. vivax* parasite?

whenever possible, would be good to mention also the temperature of the patients.

There are 2 parts of the results with different heading referring to Fig 1. I'd combine the text, or make two independent Figures.

the filtering used to consider the genes incited in the DE analyses should be further explained. I believe filterByExpr will include genes having a minimum count of reads in a certain number samples. whatever that minimum and that certain number is should be clearly stated in the methods, and mentioned in the results, so that the reader can better judge the data.

When describing Figs EV2E-G it should be made clear if these data is a comparison of the symptomatic Pv infected data presented in the current manuscript with the Pf infected data reported in the Studniberg et al., 2022 article

Fig 3Bi may be a mistake and instead be Fig 3B.

When describing Fig 4 the authors mention that "many genes downregulated by symptomatic *P. vivax* malaria remained downregulated during sub-clinical *P. vivax* infections" I suggest rewording to avoid the word remained, as this may lead the reader to perceive a longitudinal analyses, where the sub-clinical *P. vivax* infection followed an earlier symptomatic state.

It is unclear to me if Table EV1 contains new data or is just reproducing the finding in Ioannidis et al., 2021. I'd clarify and keep table EV1 only if the data is new. Additionally there seems to be no stats.

Fig 5 D-H should have some color indication of the module represented to help the reader.

One the 2nd to last paragraph of page 13 the authors mention that the fig 5 H is consistent with the findings for asymptomatic malaria in Figs 2 and 3, and it is true, but is a bit of circular syllogism as the data is all the same, so it is showing only that the modules are well defined. I'd reword.

Fig. 6 B and F have similar axes legends, I suggest clarifying the distinction, as stated in the fig legend.

I recommend adding line number to the manuscript to help the reviewers guide their comments.

Reviewer #2:

General remarks

Studniberg et al. have submitted a manuscript where they investigated the transcriptomic profile of individuals with symptomatic and asymptomatic *P. vivax* infection and non-infected controls from Papua, Indonesia. At the later stages of the ms they also correlated their transcriptomic data with a previously published mass-cytometry dataset from the same cohort and with clinical data. The main conclusions are that *P. vivax* infection seems to drive an immunoregulatory response associated with the upregulation of checkpoint inhibitor genes among T cells (especially among symptomatic donors) and reduced expression of proinflammatory pathways associated with monocytes (among both infected groups), leading to their main conclusion that monocyte dysregulation is a key feature of *P. vivax* infections.

The majority of the analysis in the study (except the first and last figure) is based on transcriptomic data and is connected to specific cell types and functions via database associations (KEGG, IPA, GO), literature review, and cell-type deconvolution. There were no functional or confirmatory experiments to verify/validate the transcriptomic data.

The study refers to and leans quite heavily on two other studies from the group (Ioannidis et al. JCI insight 2021 and Studniberg et al. MSB 2022). In Ioannidis et al 2021, they performed mass cytometry to assess B and T cell subsets in overlapping *P. vivax* donors with some of the data also used in this study. In Studniberg et al. they investigated *P. falciparum* responses in a manner similar to this study and the Ioannidis study combined but also include functional confirmatory experiments in mice. The manuscript structure and conclusions are also quite similar. I think it could be of value to also discuss the results more in context with other studies, such as (PMID: 37616070) which used a systems immunology approach to compare pf and pv immune responses, in CHMI. There is also a selected discussion on the role of CSF1 for example, but a more significant candidate in the same analysis was p53, which is not mentioned much but has been indicated in another malaria study, which was not mentioned but could be interesting to discuss (PMID: 31492649).

The cohort that is used is well defined and the comparison between clinical, asymptomatic and healthy controls is interesting. Using almost only transcriptomic data in this study lends less weight to the conclusions and I recommend confirmatory data on the monocytes. Especially since these were not further investigated in the Ioannidis 2021 study or the Studniberg 2022 study.

Although I find the bioinformatic analysis of the manuscript robust, I think that using only cell deconvolution and bulk transcriptomic data querying databases is quite limiting. Partly because the large amounts of data necessitates a selection of what is discussed or investigated closer, easily ending up fitting into a specific narrative which is not confirmed experimentally.

Major points

In the abstract, it is stated that Th2 bias and class-switch memory reduce risk of symptomatic disease. On page 14 in the results, the association is with healthy controls. I find it problematic that the authors consider the healthy control group as protected from disease and having an immune response to strive for. No information has been provided that suggests that these individuals are more protected than others (such as longitudinal data registering infections or some such). From my point of view, the main comparison should be between the two infected groups, as they can provide information on how the infection is controlled. The current study cannot address susceptibility to infection.

I don't agree with the term dysfunctional, which is extensively used in the discussion of the monocytes. In my mind, something that potentially leads to human survival (since it's the clinical disease that is associated with lethal symptoms) is not dysfunctional, but perhaps rather "modulated" to be less responsive to stimulation/activation. This is just my thoughts and preference though.

The title and abstract are somewhat misleading. 3/4 of the study is based on whole-blood transcriptomics, not integrated. Would recommend a change to remove systems and dysfunctional, and maybe include transcriptomic and modulation.

The way the results for Figure 5 are written, it's easy to interpret it as if the CyTOF data was included in the WGCNA analysis to

provide an integrated analysis. Please make it more clear that it is only included in the later spearman correlation analysis.

Regarding part "comprehensive picture of immune response" - correlation-based graph, more methodological description needed. Is the network based on the raw spearman correlation coefficient? If so, how are negative correlations handled? To have direction-independent weight, transformation into distance, e.g. similar to Euclidian distance is usually used. Unfortunately, I have difficulties following the authors argumentation on how the presented network supports the hypothesis that "high parasitemia levels during symptomatic infection drive a proliferative transcriptional profile supporting the expansion of T-bet+ MBCs and TFH cells to help control acute infection"? What is that strong association based on? The centrality of a given node? Visual inspection? Community detection?

For the last figure, the authors aim to identify predictors of disease susceptibility. I think this reasoning is problematic because they compare infected groups with a non-infected group. There is no data shown that the non-infected are more or less susceptible. What we know is that they are simply not infected at this time. Therefore, the predictor is rather about signatures associated with infection. Not predictors or susceptibility. It's also worth to note that the auc of the result is quite low, so I would not consider this a strong/robust signature of infection. The follow-up analysis of the selected genes is interesting though.

Minor points

The work presents present work relies heavily on bioinformatic analysis, mainly performed in r. The authors should provide the code for their analysis (e.g. via a open-source platform such as GitHub) to make the analysis transparent and easier to evaluate.

In the abstract, the sentence "Despite allowing...asymptomatic individuals" is important but unclear.

In the abstract, I think calling dysregulation a "critical" feature of Pv malaria is too strong a statement. It could be a hallmark, or key feature of the transcriptional profile of Pv malaria..

In the introduction and discussion, part of the narrative is promoted quite strongly that asymptomatic infections might be detrimental and suppress other immune responses, such as those to malaria vaccines. However, there is no mentioning if study participants in those studies were treated or tested for malaria (which I assume would be commonly done) or the potential impact of pre-existing antibodies (which I find more likley). Perhaps there are epidemiological studies that would support a more general immune-dysfuntion hypothesis, such as evaluation of tetanus, measles etc vaccine studies in endemic areas?

The result part around Figure 2/3 - describes many metabolic changes(up), what does that mean? The authors could try to contextualise more. Similar, in symptomatic vivax they observe upregulation of cell death, proteasomal catabolic processes.

Page 9 "...upregulation of pathways involved in cell death receptor signalling and IL-10 signalling (PDCD1, PDCD2, and TLR10)". Should TLR10 be IL10?

The asymptomatic group is sometimes referred to with different names, such as clinically silent etc. Make sure consistent.

In figure 5I, there are also monocyte nodes etc that I don't think are from the Ioannidis 2021 papers. Where are those coming from? Inferred data?

Presentation and style

Please include line numbers and figure numbers within the figure. It makes it easier to review.

All figure panels require some work-over to make the figures more size-adjusted to each other. The current figure panels are busy and feel unstructured.

Figure 1 - A-F should be smaller

Consider switching A and B in Figure2/3

The authors use DEGs to perform three different methods of gene enrichment tools and see overlapping results. I would suggest to only present one in the main figure and appending the other to supplementary

In Figure 5, when showing 6 heatmaps of similar appearance, please make it more obvious what WGCNA module they belong to. It would also be fine to move them to supplementary, as it was difficult to get much information from them.

Figure 5/6, heatmaps and network needs a legends (especially 5I had many different colors etc that needs further clarity).

Reviewer #3:

This is a very interesting paper showing evidence to support that persistent asymptomatic *P. vivax* malaria can significantly regulate the physiology of circulating blood monocytes.

Generally, the manuscript is well written and the experimental approaches are sound and well justified. I do have, however, some concerns regarding interpretation of the results and a few minor suggestions.

Major points:

- 1- It is well established that age can impact phenotype and frequencies of blood monocytes (e.g. Ryan G. Snodgrass 2022, Cao 2022). Authors show that the AM cohort they study is younger on average as compared to HC and SM. I would be important for authors to highlight this potential confounding factor.
- 2- Throughout the manuscript, authors insist on a dysfunction of monocyte in both AM and SM. However, authors do not directly evaluate monocyte function in this study (e.g. by measuring ex vivo phagocytic capacities, antigen presentation, cytokine production, etc.) Therefore, I would rather suggest an alternative more faithful terminology for their discoveries, e.g. "monocyte transcriptome deregulation"
- 3- Page 10, authors conclude their data suggest "dysregulation of monocyte effector function" in SM. They further state in page 15 that "monocyte dysfunction might be a key feature of *P. vivax* malaria." However, authors do not directly evaluate function of monocytes in this study. The results that authors obtained could also be explained by a drop in the frequencies of specific subsets of monocytes in circulation, e.g. as a consequence of monocytes being recruited out of circulation and into specific tissues to combat parasite sequestration. This, in turn, would lower the representation of genes associated with monocytes in samples from SM and AM. Is the proportion of circulating classical and intermediate monocytes significantly lower in SM and/or AM as compared to HC? Can the authors provide flow cytometry data to answer this question? Alternatively, can authors further subdivide the monocyte population in Fig 1I into different subpopulations of monocytes?
- 4- fig 3 H and I. Is there any chance that author could generate some flow cytometry validation data to support these results? I understand the huge complexity of securing these kinds of clinical samples, but some flow validation here would greatly strengthen authors' conclusions.

Minor points:

- 5- Line numbers on the manuscript would be greatly appreciated.
- 6- Please, check for typos and spaces throughout the manuscript.
- 7- Abstract phrase "Despite allowing transcriptional profiles supporting T cell differentiation, a TH2 cell bias and class-switched memory B cells reducing risk of symptomatic infection, monocyte dysfunction persisted in asymptomatic individuals." Is a bit unclear. Please consider re-phrasing this.
- 8- Last phrase on page 3, separate into two different phrases for clarity?
- 9- Consider separating study cohort characteristics from first lot of transcriptomics analysis i.e. separate fig 1A-F from 1G-J.
- 10- Fig 1G, Is this a principal component analysis (PCA) plot? If so, please clarify on text and figure legend.
- 11- Fig 3, there are several cell populations which are shared between H and I, e.g. naïve B cells, non-classical monocytes, NK cells, Plasmacytoid DCs - Does this mean that these populations are increased, decreased or remain at similar levels when comparing SM with HC? Please, clarify.
- 12- Page 10, BTM, please consider not to use acronyms for this term. It is not a broadly recognised term, and forces to interrupt reading flow to go back and check the meaning of the term.
- 13- Fig 4, legends for A, B and C are mixed up in both the main text and the figure legend.
- 14- Fig 4D and E, did authors mean AM instead of SM? Please, double check.

We sincerely would like to thank the reviewers for taking the time to assess our manuscript. Systems papers are not easy to review, and our study has been greatly improved by their input.

Reviewer #1:

Studniberg et al present a thorough analysis of transcriptional profiles of PBMCs collected from symptomatic and asymptomatic individuals infected with P. vivax alongside those of healthy controls from Papua in Indonesia.

The data is well explained and will be of interest to parasite immunologists, I suggest however a few points where I believe a more clearly could be afforded to the reader.

I believe the abstract misleads the reader to think that mass cytometry was done in the contact of this study, and it should be clarified that the data used has already been published. Also in the abstract there is unclear sentence "Despite allowing transcriptional profiles supporting T cell differentiation, a TH2 cell bias and class-switched memory B cells reducing risk of symptomatic infection, monocyte dysfunction persisted in asymptomatic individuals." I suggest rewording.

We thank the reviewer for pointing out these issues with the abstract. We have amended the abstract clarifying that mass cytometry has been previously reported. We have also reworded the unclear sentence (lines 36-39), reflecting more clearly the main results reported in the current study rather than previously published observations.

Throughout the manuscript several sentences are very long and hard to follow. Between page 2 and 3 is a good example that could be easily cut into two to help the reader.

We have edited this sentence for clarity (lines 69-72) and have checked the remainder of the manuscript for brevity.

The first sentence of page 3 requires a reference, and should specifically mention if referring to Pv or Pf malaria.

We thank the reviewer for this observation and have felt it most appropriate to remove the sentence.

The last sentence of the introduction needs a reference to tell the reader which data are the findings consistent with.

Our previous study has now been cited (line 102).

In the beginning of the results there is an unnecessary the repetition of the reference of the cohort. Saying once that it was already reported will suffice.

We have deleted the second reference.

I suggest indicating the number of study participants analysed in the text of the result section, as well as the age range.

We have now added this information to the text of the results section for clarity (lines 112-115) The age range for the study is shown in Figure 1D and EV1D.

Fig 1F should have represented a line of something to indicate the negative control IgG level. How was it assessed that all individuals in the healthy control group exhibited previous exposure to the P. vivax parasite?

As indicated in our previous publication (Ioannidis, JCI insight 2021) Australian unexposed controls were included in the evaluation of antibody responses to DBP. We have now added a dotted line illustrating background levels of Melbourne unexposed controls.

Whenever possible, would be good to mention also the temperature of the patients.

Temperature of the participants was part of the selection criteria. That information is described in the methods section.

There are 2 parts of the results with different heading referring to Fig 1. I'd combine the text, or make two independent Figures.

We agree with the reviewer and have split the figures up into two.

The filtering used to consider the genes incited in the DE analyses should be further explained. I believe filterByExpr will include genes having a minimum count of reads in a certain number samples. whatever that minimum and that certain number is should be clearly stated in the methods, and mentioned in the results, so that the reader can better judge the data.

We appreciate the request for clarification by the reviewer. The filterByExpr default function retains genes with a minimum of 10 counts-per-million reads in the minimum number of samples per group. We have rephrased the methods for clarity (lines 633-634).

When describing Figs EV2E-G it should be made clear if these data is a comparison of the symptomatic Pv infected data presented in the current manuscript with the Pf infected data reported in the Studniberg et al., 2022 article

We have now amended the sentence to make it clear we are comparing transcriptional profiles from this study with the 2022 study (lines 245-250).

Fig 3Bi may be a mistake and instead be Fig 3B

We thank the reviewer for noticing this error and have changed the text to read as Fig 3B (line 192).

When describing Fig 4 the authors mention that "many genes downregulated by symptomatic P. vivax malaria remained downregulated during sub-clinical P. vivax infections" I suggest rewording to avoid the word remained, as this may lead the reader to perceive a longitudinal analyses, where the sub-clinical P. vivax infection followed an earlier symptomatic state.

The reviewer raises a valid point. We have replaced the word “remained” with “found to be” (line 236)

It is unclear to me if Table EV1 contains new data or is just reproducing the finding in Ioannidis et al., 2021. I'd clarify and keep table EV1 only if the data is new. Additionally there seems to be no stats.

Thanks for the observation. We have clarified that these data were previously published and have removed the table.

Fig 5 D-H should have some color indication of the module represented to help the reader.

We have added in module names to the figure.

One the 2nd to last paragraph of page 13 the authors mention that the fig 5 H is consistent with the findings for asymptomatic malaria in Figs 2 and 3, and it is true, but is a bit of circular syllogism a the data is all the same, so it is showing only that the modules are well defined. I'd reword.

We appreciate the reviewer’s suggestion and acknowledge that there is room for clarification here. The WGCNA pipeline takes the filtered and normalised expression values from the whole genome prior to differential expression analysis. Therefore, WGCNA module genes will contain not only the genes determined as differently expressed from the limma-voom differential expression pipeline, but also all those genes that were deemed insignificant by the limma-voom analysis. Due to this, the tan module does not contain the same genes as those differentially expressed in the AMvHC contrast. Because these are two different datasets generated from two different analysis pipelines, we deem it appropriate to describe results of the gene ontology analysis using WGCNA gene modules as consistent with that from the gene ontology analysis using DE genes between AMvHC. We have amended the text for clarity (lines 320-321). We have also included a table legend for Table 1 so that the origin of the gene sets used for the gene ontology analysis were clarified.

Fig. 6 B and F have similar axes legends, I suggest clarifying the distinction, as stated in the fig legend.

We have now clarified that whereas Fig B depicts predicted frequencies using genes downregulated by symptomatic *P. vivax*, Fig F shows frequencies using genes downregulated by both symptomatic and asymptomatic infection (lines 3432-346). This is also illustrated in the figure axes. The figure legend has been further clarified to specify the number of genes submitted for logistic modelling.

I recommend adding line number sot the manuscript to help the reviewers guide their comments.

We appreciate the reviewer pointing this out. We have added line numbers and will ensure inclusion of line numbers for future submissions.

Reviewer #2:

General remarks

Studniberg et al. have submitted a manuscript where they investigated the transcriptomic profile of individuals with symptomatic and asymptomatic P. vivax infection and non-infected controls from Papua, Indonesia. At the later stages of the ms they also correlated their transcriptomic data with a previously published mass-cytometry dataset from the same cohort and with clinical data. The main conclusions are that P. vivax infection seems to drive an immunoregulatory response associated with the upregulation of checkpoint inhibitor genes among T cells (especially among symptomatic donors) and reduced expression of proinflammatory pathways associated with monocytes (among both infected groups), leading to their main conclusion that monocyte dysregulation is a key feature of P. vivax infections.

The majority of the analysis in the study (except the first and last figure) is based on transcriptomic data and is connected to specific cell types and functions via database associations (KEGG, IPA, GO), literature review, and cell-type deconvolution. There were no functional or confirmatory experiments to verify/validate the transcriptomic data.

The study refers to and leans quite heavily on two other studies from the group (Ioannidis et al. JCI insight 2021 and Studniberg et al. MSB 2022). In Ioannidis et al 2021, they performed mass cytometry to assess B and T cell subsets in overlapping P. vivax donors with some of the data also used in this study. In Studniberg et al. they investigated P. falciparum responses in a manner similar to this study and the Ioannidis study combined but also include functional confirmatory experiments in mice. The manuscript structure and conclusions are also quite similar. I think it could be of value to also discuss the results more in context with other studies, such as (PMID: 37616070) which used a systems immunology approach to compare pf and pv immune responses, in CHMI. There is also a selected discussion on the role of CSF1 for example, but a more significant candidate in the same analysis was p53, which is not mentioned much but has been indicated in another malaria study, which was not mentioned but could be interesting to discuss (PMID: 31492649).

The cohort that is used is well defined and the comparison between clinical, asymptomatic and healthy controls is interesting. Using almost only transcriptomic data in this study lends less weight to the conclusions and I recommend confirmatory data on the monocytes. Especially since these were not further investigated in the Ioannidis 2021 study or the Studniberg 2022 study.

Although I find the bioinformatic analysis of the manuscript robust, I think that using only cell deconvolution and bulk transcriptomic data querying databases is quite limiting. Partly because the large amounts of data necessitates a selection of what is discussed or investigated closer, easily ending up fitting into a specific narrative which is not confirmed experimentally.

We thank the reviewer for the fair appraisal of our study. In this revised version we have incorporated major changes to provide a more balanced delivery of the conclusions, including discussion of the aforementioned studies in CHMI (lines 493-496), provision of new cytokine secretion functional data (lines 237-43) and extensive phenotypic characterisation of the blood monocyte compartment by mass cytometry.

Major points

In the abstract, it is stated that Th2 bias and class-switch memory reduce risk of symptomatic disease. On page 14 in the results, the association is with healthy controls. I find it problematic that the authors consider the healthy control group as protected from disease and having an immune response to strive for. No information has been provided that suggests that these individuals are more protected than others (such as longitudinal data registering infections or some such). From my point of view, the main comparison should be between the two infected groups, as they can provide information on how the infection is controlled. The current study cannot address susceptibility to infection.

We appreciate the reviewer's observation. We should point out that following recommendations of other reviewers, the abstract has been largely rephrased to reflect more the results obtained in the current rather than our previous mass cytometry work.

I could not agree more with the reviewer in that longitudinal cohorts are required to fully demonstrate a protective effect over time. However cross-sectional cohorts like the one used in the peer reviewed study for estimation of B cell and T cell mass cytometry phenotypes (Ioannidis et al, 2021) allowed calculation of odds ratios using logistic regression, which provide a validated tool to evaluate odds or risk that of an event happening or not. We revised the manuscript to use this language, avoid overclaiming results and reflect exactly the test that was used to evaluate our previous data. The text in page 14 has been modified.

I don't agree with the term dysfunctional, which is extensively used in the discussion of the monocytes. In my mind, something that potentially leads to human survival (since it's the clinical disease that is associated with lethal symptoms) is not dysfunctional, but perhaps rather "modulated" to be less responsive to stimulation/activation. This is just my thoughts and preference though.

We thank the reviewer for sharing their point of view. Our new mass cytometry results identified clear differences in the composition of the monocyte compartment occurring in response to infection. Deletion of specific monocyte subsets appear to be associated with the reduced monocyte transcriptional signature reported in our first submission. This is now reflected throughout the discussion and the term dysfunction has been replaced on various sentences to reflect our new findings and to provide a more balanced discussion of our results.

The title and abstract are somewhat misleading. 3/4 of the study is based on whole-blood transcriptomics, not integrated. Would recommend a change to remove systems and dysfunctional, and maybe include transcriptomic and modulation.

In light of the addition of other endpoints including new cytokine assays and CyTOF data, we feel that is now appropriate use the term "systems" is the title of the study. As per our comment above, our new data lend support that *P. vivax* malaria might result in an imbalance

of the composition of the monocyte compartment, with downstream functional implications. This has been noted in the title and abstract.

The way the results for Figure 5 are written, it's easy to interpret it as if the CyTOF data was included in the WGCNA analysis to provide an integrated analysis. Please make it more clear that it is only included in the later spearman correlation analysis.

The entire CyTOF data set showing significant changes in odds of infection was in fact included in the data set. Due to space constrains, the WGCNA heatmap shown in figure 6 was a small subset of the analysis highlighting the gene modules that show correlation with clinical groups. We thank the reviewer for pointing out this was not clear. The full matrix has now been included in the expanded view section.

Regarding part "comprehensive picture of immune response" - correlation-based graph, more methodological description needed. Is the network based on the raw spearman correlation coefficient? If so, how are negative correlations handled? To have direction-independent weight, transformation into distance, e.g. similar to Euclidian distance is usually used. Unfortunately, I have difficulties following the authors argumentation on how the presented network supports the hypothesis that "high parasitemia levels during symptomatic infection drive a proliferative transcriptional profile supporting the expansion of T-bet+ MBCs and TFH cells to help control acute infection"? What is that strong association based on? The centrality of a given node? Visual inspection? Community detection?

The network is based on the raw Spearman correlation coefficient. Correlations in the network visualisation are handled in a direction-dependent manner, whereby nodes connected by blue lines are negatively correlated and nodes connected by red lines are positively correlated. The distance between nodes is calculated via a prefuse force-directed layout based on correlation strength, as per the force-directed paradigm described by Heer *et al* (Heer et al, 2005). We have added more information in the text to help interpretation of our network (lines 348-349 and 352-353).

For the last figure, the authors aim to identify predictors of disease susceptibility. I think this reasoning is problematic because they compare infected groups with a non-infected group. There is no data shown that the non-infected are more or less susceptible. What we know is that they are simply not infected at this time. Therefore, the predictor is rather about signatures associated with infection. Not predictors or susceptibility. It's also worth to note that the auc of the result is quite low, so I would not consider this a strong/robust signature of infection. The follow-up analysis of the selected genes is interesting though.

We appreciate the reviewer's comment and agree that these genes do not represent predictors of disease susceptibility. Our intention in this analysis was to identify the most important genes and pathways modulating monocyte function in symptomatic or asymptomatic *P. vivax* malaria in this cohort. As such, we have reworded the results section (lines 346-348 and 363-365) for this figure for clarity.

Minor points

The work presents present work relies heavily on bioinformatic analysis, mainly performed in R. The authors should provide the code for their analysis (e.g. via a open-source platform such as GitHub) to make the analysis transparent and easier to evaluate.

Whilst it is not a requirement of the journal, we are working on making the source code available on GitHub.

In the abstract, the sentence "Despite allowing...asymptomatic individuals" is important but unclear.

We have edited the sentence for clarity (lines 36-39).

In the abstract, I think calling dysregulation a "critical" feature of Pv malaria is too strong a statement. It could be a hallmark, or key feature of the transcriptional profile of Pv malaria.

We appreciate the reviewer's opinion here and have re-worded the statement (lines 39-40).

In the introduction and discussion, part of the narrative is promoted quite strongly that asymptomatic infections might be detrimental and suppress other immune responses, such as those to malaria vaccines. However, there is no mentioning if study participants in those studies were treated or tested for malaria (which I assume would be commonly done) or the potential impact of pre-existing antibodies (which I find more likely). Perhaps there are epidemiological studies that would support a more general immune-dysfunction hypothesis, such as evaluation of tetanus, measles etc vaccine studies in endemic areas?

We thank the reviewer for their feedback. A study published late 2024 (Diawara et al, 2024) found that the PfSPZ vaccine was efficacious only when individuals had received presumptive antimalarial treatment prior to first vaccine dose. We have referenced this study in the introduction (lines 86-90).

The result part around Figure 2/3 - describes many metabolic changes(up), what does that mean? The authors could try to contextualise more. Similar, in symptomatic vivax they observe upregulation of cell death, proteasomal catabolic processes.

Thank you for the comment. We have now addressed this finding in the context of our previous observations that show increased fatty acid metabolism transcripts to be increased in *P. falciparum* malaria (Studniberg *et al*, 2022), which suggests that enhanced metabolism may be a result of active, clinical malaria infection (lines 159-162).

Page 9 "...upregulation of pathways involved in cell death receptor signalling and IL-10 signalling (PDCD1, PDCD2, and TLR10)". Should TLR10 be IL10?

We appreciate the reviewer picking up on this mistake. We have fixed this.

The asymptomatic group is sometimes referred to with different names, such as clinically silent etc. Make sure consistent.

Clinically silent has been replaced with asymptomatic throughout the manuscript.

In figure 5I, there are also monocyte nodes etc that I don't think are from the Ioannidis 2021 papers. Where are those coming from? Inferred data?

These monocyte nodes are estimated cell frequencies from cell-type deconvolution as described in the methods. We have included a key within this figure for clarity.

Presentation and style

Please include line numbers and figure numbers within the figure. It makes it easier to review.

Line number have been included in the revised version of the manuscript.

All figure panels require some work-over to make the figures more size-adjusted to each other. The current figure panels are busy and feel unstructured.

We thank the reviewer for this feedback and have improved the scaling of the figure panels.

Figure 1 - A-F should be smaller

Figures A-F have been moved to their own figure and so are in proportion now.

Consider switching A and B in Figure2/3

We have taken the reviewer's suggestion on board and have switched figures 3/4 A and B.

The authors use DEGs to perform three different methods of gene enrichment tools and see overlapping results. I would suggest to only present one in the main figure and appending the other to supplementary

We have taken the reviewer's suggestion on board and have moved the KEGG pathway analysis to supplementary but feel that the GO and Ingenuity pathway analyses must remain in the main figure as they provide complimentary information.

In Figure 5, when showing 6 heatmaps of similar appearance, please make it more obvious what WGCNA module they belong to. It would also be fine to move them to supplementary, as it was difficult to get much information from them.

We thank the reviewer for their observation and agree that additional information was required for ease of comprehension. We have included the WGCNA module name above each heatmap for clarity.

Figure 5/6, heatmaps and network needs a legends (especially 5I had many different colors etc that needs further clarity).

We have included legends for these figures now.

Reviewer #3:

*This is a very interesting paper showing evidence to support that persistent asymptomatic *P. vivax* malaria can significantly regulate the physiology of circulating blood monocytes. Generally, the manuscript is well written, and the experimental approaches are sound and well justified. I do have, however, some concerns regarding interpretation of the results and a few minor suggestions.*

We thank the reviewer for the positive appraisal of our manuscript.

Major points:

1- It is well established that age can impact phenotype and frequencies of blood monocytes (e.g. Ryan G. Snodgrass 2022, Cao 2022). Authors show that the AM cohort they study is younger on average as compared to HC and SM. I would be important for authors to highlight this potential confounding factor.

Thank you for the fair comment. Our differential expression analysis pipeline involved a robust pipeline for adjustment for confounding variables. Ultimately, we found the RUVr method to be optimal for the detection and adjustment for surrogate variables, allowing us to capture the majority of known or unknown variation. We observed correlations between the first 10 principal components, clinical variables, and surrogate variables to ensure that significant variation was adjusted for. From this approach, we determined that adjustment for 5 surrogate variables was optimal for capturing all variation up to PC8. Interestingly, age did not significantly correlate with any principal components. We have adjusted the methods describing our RUVr analysis to specifically state that significant known and unknown variation was removed. A statement has been included in the Results section (lines 134-136) to clarify our approach for adjustment of confounding variables.

2- Throughout the manuscript, authors insist on a dysfunction of monocyte in both AM and SM. However, authors do not directly evaluate monocyte function in this study (e.g. by measuring ex vivo phagocytic capacities, antigen presentation, cytokine production, etc.) Therefore, I would rather suggest an alternative more faithful terminology for their discoveries, e.g. "monocyte transcriptome deregulation"

We have now included new functional cytokine secretion data that is consistent with our transcriptional profiling and shows reduced IL-1-b and TNF secretion in isolated monocytes from *P. vivax*-infected study participants compared to those from healthy controls (lines 238-244). We still appreciate the reviewer's perspective and agree in that the main endpoint evaluated in the study is transcriptomics. Thus, where appropriate, we have reworded the manuscript to "monocyte imbalance" or monocyte transcriptome regulation.

3- Page 10, authors conclude their data suggest "dysregulation of monocyte effector function" in SM. They further state in page 15 that "monocyte dysfunction might be a key feature of *P. vivax* malaria." However, authors do not directly evaluate function of monocytes in this study. The results that authors obtained could also be explained by a drop in the frequencies of specific subsets of monocytes in circulation, e.g. as a consequence of monocytes being recruited out of circulation and into specific tissues to combat parasite sequestration. This, in turn, would lower the representation of genes associated with monocytes in samples from SM and AM. Is the proportion of circulating classical and intermediate monocytes significantly lower in SM and/or AM as compared to HC? Can the authors provide flow cytometry data to answer this question? Alternatively, can authors further subdivide the monocyte population in Fig 11 into different subpopulations of monocytes?

Our revised version of the study includes a full new figure (Figure 8), in which we performed high-dimensional mass cytometry to examine monocyte composition in this cohort. Using a panel of ~20 metal-labelled antibodies we were able to identify ~12 monocyte subsets in the blood. Our main findings revealed that the reduction of monocyte transcriptional activity reported in our first submission is significantly associated with the depletion of a subset of CCR2⁺CXCR4⁺ classical monocytes from the blood in *P. vivax* malaria compared to the healthy immune controls group. This new information has now been incorporated in the text and discussed extensively in the context of pre-existing literature in the Discussion session.

4- fig 3 H and I. Is there any chance that author could generate some flow cytometry validation data to support these results? In understand the huge complexity of securing these kinds of clinical samples, but some flow validation here would greatly strengthen authors' conclusions.

As per comment above, we have now included high dimensional mass cytometry to untangle the dynamics of the monocyte compartment in *P. vivax* infected individuals and healthy controls. Although sc-RNA-seq experiments which are outside the scope of this manuscript will be required to match transcriptional profiles with cell phenotypes, our data is consistent with the preferential depletion of subsets of classical monocytes aligning with a reduction in monocyte transcriptional activity observed in our RNA-seq analysis. These new data are in agreement with the deconvolution analysis provided in old figure 3.

Minor points:

5- Line numbers on the manuscript would be greatly appreciated.

We appreciate the reviewer pointing this out and line numbers are now included in the revised version of the manuscript.

6- Please, check for typos and spaces throughout the manuscript.

We thank the reviewer for pointing this out. We have had another pass through the manuscript to correct spelling and space errors.

7- Abstract phrase "Despite allowing transcriptional profiles supporting T cell differentiation, a TH2 cell bias and class-switched memory B cells reducing risk of

symptomatic infection, monocyte dysfunction persisted in asymptomatic individuals." Is a bit unclear. Please consider re-phrasing this.

We appreciate the reviewer's suggestion and have re-written this sentence, reflecting the main findings relevant to the current study for asymptomatic individuals (lines 36-39).

8- Last phrase on page 3, separate into two different phrases for clarity?

We have separated out this sentence as per the reviewer's suggestion (lines 69-72).

9- Consider separating study cohort characteristics from first lot of transcriptomics analysis i.e. separate fig 1A-F from 1G-J.

We appreciate this feedback from the reviewer, along with reviewer 1. We agree with both reviewers and have separated the study cohort characteristics from the transcriptomics analysis.

10- Fig 1G, Is this a principal component analysis (PCA) plot? If so, please clarify on text and figure legend.

Figure 2A (previously 1G) depicts a multi-dimensional scaling (MDS) plot generated by using the plotMDS function in limma. The plotMDS function uses multi-dimensional scaling to plot differences in expression profiles between samples in the analysis, in a similar manner to a PCA. We have amended the figure legend to explain this analysis more clearly.

11- Fig 3, there are several cell populations which are shared between H and I, e.g. naïve B cells, non-classical monocytes, NK cells, Plasmacytoid DCs - Does this mean that these populations are increased, decreased or remain at similar levels when comparing SM with HC? Please, clarify.

Figures 3 H-I (now figures 4 H-I) do not give information on population frequencies. They depict estimations of the proportion of cell-type marker genes that are differentially expressed and either upregulated (figure 4H) or downregulated (figure 4I) in SMvHC. This is a validated bioinformatic approach commonly used to infer cellular sources of transcriptional signatures in bulk RNA-seq experiments. We have extended the explanation in the text to make that point clearer.

12- Page 10, BTM, please consider not to use acronyms for this term. It is not a broadly recognised term, and forces to interrupt reading flow to go back and check the meaning of the term.

We have removed the acronyms for clarity and ease of reading.

13- Fig 4, legends for A, B and C are mixed up in both the main text and the figure legend.

We are very grateful for the reviewer picking up on this error. We have reordered figures 4A-C so that they read in-line with the text and figure legend.

14- Fig 4D and E, did authors mean AM instead of SM? Please, double check.

Please note Figs. 4D and E are now Figs. 5D and E.

We appreciate the reviewer pointing this out. Upon re-evaluation, the text was written correctly. However, we appreciate this could have been clearer, and have edited the text for clarity (lines 283-285).

1st Jul 2025

Manuscript Number: MSB-2024-12610R

Title: Systems approach identifies monocyte imbalance in symptomatic and asymptomatic *P. vivax* malaria

Dear Prof Hansen,

Thank you for the submission of your revised manuscript to Molecular Systems Biology. I am pleased to inform you that we will be able to accept your manuscript pending the following final amendments and appropriate response to reviewers:

- 1) Please upload your source code for transcriptomic analysis to a suitable repository such as Github to ensure that it is freely publicly available. Please also include a README file with practical use instructions for potential future users of your code.
- 2) Please format the Data Availability section according to the example below:
"The datasets and computer code produced in this study are available in the following databases:
- Chip-Seq data: Gene Expression Omnibus GSE46748 (<https://www.ncbi.nlm.nih.gov/geo/query/acc.cgi?acc=GSE46748>)
- Modeling computer scripts: GitHub (<https://github.com/SysBioChalmers/GECKO/releases/tag/v1.0>)
- [data type]: [full name of the resource] [accession number/identifier] ([doi or URL or identifiers.org/DATABASE:ACCESSION])"
- 3) Our journal encourages inclusion of *data citations in the reference list* to directly cite datasets that were re-used and obtained from public databases. Data citations in the article text are distinct from normal bibliographical citations and should directly link to the database records from which the data can be accessed. In the main text, data citations are formatted as follows: "Data ref: Smith et al, 2001" or "Data ref: NCBI Sequence Read Archive PRJNA342805, 2017". In the Reference list, data citations must be labeled with "[DATASET]". A data reference must provide the database name, accession number/identifiers and a resolvable link to the landing page from which the data can be accessed at the end of the reference. Further instructions are available at .
- 4) In the Methods, please ensure that a statement on whether or not blinding was done is included in the Methods even if no blinding was done. Please also be sure to update the Author Checklist with this information and where it can be found in the manuscript.
- 5) When submitting your revised manuscript, please do not include the Reagents and Tools Table in the Methods section of the manuscript but upload it as a separate file choosing the file type "Reagent Table".
- 6) Please place individual sections of the manuscript in the following order: Title page - Abstract & Keywords - Introduction - Results - Discussion - Methods - Data Availability - Acknowledgements - Disclosure and Competing Interests Statement - References - Figure Legends - Expanded View Figure Legends.
- 7) For the figures and figure legends, please take care of the following:
 - Please note that the exact p values are not provided in the legends of figures 1D, E; EV1 C, E; EV6A, B.
 - Please indicate the statistical test used for data analysis in the legends of figures 2D, 3C, D, E; 4C, D; 5A, D; EV2, EV3, EV4C, F, G; EV5, EV7 A.
 - Please note that the box plots need to be defined in terms of minima, maxima, centre, bounds of box and whiskers, and percentile in the legends of figures 7B, F.
 - Please note that the box plots need to be defined in terms of bounds of box and whiskers, and percentile in the legends of figures 8D-F.
- 8) Expanded View figure and tables should be uploaded as individual high-resolution files. Alternatively, the single PDF file that you have uploaded can be renamed the Appendix file. In that case please ensure that the title page contains "Appendix for + manuscript title" and a Table of Contents with the page numbers for the listed items; the nomenclature should be Appendix Figure Sx and Appendix Table Sx throughout the manuscript and Appendix PDF, and legends should then be moved from the main manuscript file to below the relevant figure/table in the Appendix.
- 9) Please ensure that all funding sources are entered into the manuscript submission system. Currently Victorian State Government Operational Infrastructure Support is missing.
- 10) As part of the EMBO Publications transparent editorial process initiative (see our policy here: https://www.embopress.org/transparent-process#Review_Process), Molecular Systems Biology will publish online a Peer Review File (PRF) to accompany accepted manuscripts. This file will be published in conjunction with your paper and will include the anonymous referee reports, your point-by-point response and all pertinent correspondence relating to the manuscript. Let us know whether you agree with the publication of the PRF and as here, if you want to remove or not any figures from it prior to publication. Please note that the Authors checklist will be published at the end of the PRF.
- 11) After your paper is published, we may promote it on social media. If you have any handles or hashtags for Bluesky you would like included, please let us know.
- 12) Please provide a point-by-point letter INCLUDING my comments as well as the reviewer's reports and your detailed responses (as Word file).

I look forward to reading a new revised version of your manuscript as soon as possible.

Yours sincerely,

Poonam Bheda, PhD
Scientific Editor
Molecular Systems Biology

Reviewer #1:

The authors have addressed my comments, and the manuscript has significantly improved in clarity and conclusions are now more robustly supported by data and analyses. I recommend publication.

Reviewer #2:

Overall the authors have addressed our comments satisfactorily and the manuscript has greatly benefited from adding the mass cytometry dataset and toning down some of the conclusions and interpretations. There are still extensive use of the word dysregulation, which I find problematic as it's not a neutral word, but I leave up to the editor to decide on.

We do have some remaining minor comments and questions.

Abstract

-line 16 - "might not support all immune processes required to control parasitemia..." in asymptomatic parasitemia could be considered controlled, no? Maybe eliminate parasitemia?

Introduction

- Line 60-61 similar to abstract statement "Immune processes capable of fully controlling parasitemia", ref to studniberg et al 2022, maybe rather "full eliminate parasitemia"? because without symptoms, the infection seems controlled to me

Results

Healthy immune control group - Fig1E not included in parasitemia assessment. I assume they are from the same area - where they screened for parasitaemia?

Paragraph 163-215 "Symptomatic *P. vivax* malaria transcriptional profiles feature upregulation of checkpoint receptors and downregulation of monocyte effector transcripts"

o Line 192 - "in addition to the upregulation of anti-inflammatory transcripts" - which transcripts are anti-inflammatory? Is this based on upregulation of LAG3 and CTLA4? Also, transcripts per se are not inflammatory; it's the proteins they encode for

o Line 203, why CSF1? It is 4th place on the list, the first one, p53 is in the context of monocytes also relevant (Tran et al 2019)

Line 178 "...signaling and IL-10 signaling (PDCD1, PDCD2, and TLR10), as well..." We had identified that TLR10 is likely an error and it said in the rebuttal that you had corrected this. Seems not to be the case, or should it be TLR10?

Paragraph 217-257 "*P. vivax* asymptomatic malaria supports T cell differentiation but compromises the blood monocyte compartment"

- Line 239 - why stimulate with LPS when you are interested in malaria responses? Potentially add as a limitation that its not a physiological stimuli in malaria context. Would also be good to explain how it's used to assess tolerance.

Line 242 - asymptomatic vs healthy do not show significance in TNF secretion but text says "TNF production was significantly compromised in monocytes from both symptomatic and asymptomatic *P. vivax*-infected individuals relative to healthy immune controls" -> see Fig EV6

Paragraph "Weighted gene co-expression network analysis of the immune response to *P. vivax* malaria"

- Line 260-267 feel like a long intro and just describing previous study results

- Line 274-277 technical details could move to the method section

Discussion

Line 498 - "...conceivable that these cells increase in response to high parasitemia *P. vivax* to reduce the burden..." -> "...high *P. vivax* parasitemia..."

No clear limitation description

Reviewer #3:

Authors have now addressed a very thorough set of referee comments. As a result, authors have greatly strengthen the scientific case for this manuscript. Not only the message is more clear and accurate now, but also the authors have produced additional and very valuable experimental data that support their conclusions. Therefore, I believe this piece of research is now ready for publication.

I would still suggest

- 1- Please, do make the source code for transcriptomic analysis available on GitHub.
- 2- Move figure EV6 from supplementary to the main body of the manuscript.

Final point-by-point reply to editor and reviewers' comments

We would like to thank the reviewers and editor for the positive appraisal of our manuscript. Detailed reply to their final comments can be found below:

1) *Please upload your source code for transcriptomic analysis to a suitable repository such as Github to ensure that it is freely publicly available. Please also include a README file with practical use instructions for potential future users of your code.*

The code and README are now on GitHub.

2) *Please format the Data Availability section according to the example below:
"The datasets and computer code produced in this study are available in the following databases:*

- *Chip-Seq data: Gene Expression Omnibus GSE46748*

(<https://www.ncbi.nlm.nih.gov/geo/query/acc.cgi?acc=GSE46748>)

- *Modeling computer scripts: GitHub*

(<https://github.com/SysBioChalmers/GECKO/releases/tag/v1.0>)

- *[data type]: [full name of the resource] [accession number/identifier] ([doi or URL or identifiers.org/DATABASE:ACCESSION)]"*

The Data Availability section has been re-formatted.

3) *Our journal encourages inclusion of *data citations in the reference list* to directly cite datasets that were re-used and obtained from public databases. Data citations in the article text are distinct from normal bibliographical citations and should directly link to the database records from which the data can be accessed. In the main text, data citations are formatted as follows: "Data ref: Smith et al, 2001" or "Data ref: NCBI Sequence Read Archive PRJNA342805, 2017". In the Reference list, data citations must be labeled with "[DATASET]". A data reference must provide the database name, accession number/identifiers and a resolvable link to the landing page from which the data can be accessed at the end of the reference. Further instructions are available*

The relevant data citation has been inserted as per the data citation guidelines.

at <https://www.embopress.org/page/journal/17574684/authorguide#referencesformat>

4) *In the Methods, please ensure that a statement on whether or not blinding was done is included in the Methods even if no blinding was done. Please also be sure to update the Author Checklist with this information and where it can be found in the manuscript.*

We have added this statement to the end of the "statistical analysis" section in Methods and have updated the Author Checklist.

5) *When submitting your revised manuscript, please do not include the Reagents and Tools Table in the Methods section of the manuscript but upload it as a separate file choosing the file type "Reagent Table".*

The Reagents and Tools Table has been removed from the manuscript and will be uploaded as a separate file.

6) Please place individual sections of the manuscript in the following order: Title page - Abstract & Keywords - Introduction - Results - Discussion - Methods - Data Availability - Acknowledgements - Disclosure and Competing Interests Statement - References - Figure Legends - Expanded View Figure Legends.

The manuscript has been re-ordered to reflect the above.

7) For the figures and figure legends, please take care of the following:
- Please note that the exact p values are not provided in the legends of figures 1D, E; EV1 C, E; EV6A, B.

Exact p values have been provided in the figure legends listed above.

- Please indicate the statistical test used for data analysis in the legends of figures 2D, 3C, D, E; 4C, D; 5A, D; EV2, EV3, EV4C, F, G; EV5, EV7 A.

Statistical tests used for data analysis have been added into the figure legends listed above.

- Please note that the box plots need to be defined in terms of minima, maxima, centre, bounds of box and whiskers, and percentile in the legends of figures 7B, F.
- Please note that the box plots need to be defined in terms of bounds of box and whiskers, and percentile in the legends of figures 8D-F.

We have now provided further information to define the box plots.

8) Expanded View figure and tables should be uploaded as individual high-resolution files. Alternatively, the single PDF file that you have uploaded can be renamed the Appendix file. In that case please ensure that the title page contains "Appendix for + manuscript title" and a Table of Contents with the page numbers for the listed items; the nomenclature should be Appendix Figure Sx and Appendix Table Sx throughout the manuscript and Appendix PDF, and legends should then be moved from the main manuscript file to below the relevant figure/table in the Appendix.

We have submitted the EV figures as separate high-res files.

9) Please ensure that all funding sources are entered into the manuscript submission system. Currently Victorian State Government Operational Infrastructure Support is missing.

This will be rectified during submission.

10) As part of the EMBO Publications transparent editorial process initiative (see our policy here: https://www.embopress.org/transparent-process#Review_Process), Molecular Systems Biology will publish online a Peer Review File (PRF) to accompany accepted manuscripts. This file will be published in conjunction with your paper and will include the anonymous referee reports, your point-by-point response and all pertinent correspondence relating to the manuscript. Let us know whether you agree with the publication of the PRF and as here, if you want to remove or not any figures from it prior to publication. Please note that the Authors checklist will be published at the end of the PRF.

The authors agree with the publication of the PRF as is.

11) After your paper is published, we may promote it on social media. If you have any handles or hashtags for Bluesky you would like included, please let us know.

Please add the handles @DrDiHansen and @steph-stud.bsky.social and the hashtag #dhansenlab.

12) Please provide a point-by-point letter *INCLUDING* my comments as well as the reviewer's reports and your detailed responses (as Word file).

This file

Reviewer #1:

The authors have addressed my comments, and the manuscript has significantly improved in clarity and conclusions are now more robustly supported by data and analyses. I recommend publication.

We thank this reviewer for their positive appraisal of the manuscript. Our study has been greatly improved by their input.

Reviewer #2:

Overall the authors have addressed our comments satisfactorily and the manuscript has greatly benefited from adding the mass cytometry dataset and toning down some of the conclusions and interpretations. There is still extensive use of the word dysregulation, which I find problematic as it's not a neutral word, but I leave up to the editor to decide on.

We do have some remaining minor comments and questions.

Abstract

-line 16 - "might not support all immune processes required to control parasitemia..." in asymptomatic parasitemia could be considered controlled, no? Maybe eliminate parasitemia?

We have now replaced the word "control" the term "eliminate" as suggested by reviewer.

Introduction

- Line 60-61 similar to abstract statement "Immune processes capable of fully controlling parasitemia", ref to studniberg et al 2022, maybe rather "full eliminate parasitemia"? because without symptoms, the infection seems controlled to me

Thanks for the comment. The authors are not making reference to the control of clinical infection but the control of parasite replication. While it is true that in sub-clinical infections there are no febrile symptoms, parasites still replicate and survive in the host. This study and our previous paper in 2022 lend support to the concept that immunosuppressive mechanism operating in asymptomatic infections prevent FULL CONTROL of parasitemia despite the absence of clinical symptoms. Furthermore, our previous research demonstrated that CTLA-4 blockade during asymptomatic chronic parasitemia in mouse infection models, facilitate

control of infection. In light of this evidence, the authors feel that “fully control” is an appropriate way to explain this process.

Results

Healthy immune control group - Fig1E not included in parasitemia assessment. I assume they are from the same area - where they screened for parasitaemia?

Full details of our study design have been provided in the Methods section (line 540), with more extensive explanations cited in our two previous publications (Ioannidis et al, JCI insight, 2021 and Studniberg et al, Mol Sys Bio, 2022). As stated throughout those articles, healthy controls were recruited from the same area and were negative for parasites by both light microscopy and PCR.

Paragraph 163-215 "Symptomatic P. vivax malaria transcriptional profiles feature upregulation of checkpoint receptors and downregulation of monocyte effector transcripts" o Line 192 - "in addition to the upregulation of anti-inflammatory transcripts" - which transcripts are anti-inflammatory? Is this based on upregulation of LAG3 and CTLA4? Also, transcripts per se are not inflammatory; it's the proteins they encode for

We have amended the sentences.

o Line 203, why CSF1? It is 4th place on the list, the first one, p53 is in the context of monocytes also relevant (Tran et al 2019)

Our analysis in Fig 4F identified 7 different upstream regulators with similar z-activation scores. From those the authors highlighted CSF1, as it a key cytokine facilitating the proliferation, survival and fate of **monocytes and macrophages**. CSF1 seems to control functions highly aligned with the changes of transcriptional profiles observed in response to infection and at the same time, unlike some of the other upstream regulators, it is highly specific for the monocyte compartment.

Line 178 "...signaling and IL-10 signaling (PDCD1, PDCD2, and TLR10), as well..." We had identified that TLR10 is likely an error and it said in the rebuttal that you had corrected this. Seems not to be the case, or should it be TLR10?

The change was lost in a previous version of the revisions. It's IL-10 now.

Paragraph 217-257 "P. vivax asymptomatic malaria supports T cell differentiation but compromises the blood monocyte compartment"

The authors are not sure what the question is.

- Line 239 - why stimulate with LPS when you are interested in malaria responses? Potentially add as a limitation that its not a physiological stimuli in malaria context. Would also be good to explain how it's used to assess tolerance.

Thank you for the comment. Monocytes were stimulated with LPS as it is the gold-standard antigen for monocyte activation and allowed us to provide proof of concept for the reduced capacity of cells from malaria infected individuals to mount cytokine responses. Our results (largely discussed in this study and in Studniberg et al 2022) support the concept that immunosuppressive responses during malaria infections might not only impair full control of

parasitemia but also have a detrimental effect on the host's capacity to respond to vaccination and other infections, including bacterial infection. As such understanding how monocytes from malaria-infected individuals react to LPS stimulation is physiologically relevant.

Line 242 - asymptomatic vs healthy do not show significance in TNF secretion but text says "TNF production was significantly compromised in monocytes from both symptomatic and asymptomatic P. vivax-infected individuals relative to healthy immune controls" -> see Fig EV6

Thanks for the comment. We have now rephased the text.

Paragraph "Weighted gene co-expression network analysis of the immune response to P. vivax malaria"
- *Line 260-267 feel like a long intro and just describing previous study results*
- *Line 274-277 technical details could move to the method section*

The CyTOF populations described in lines 260-267 are an integral part of the analysis in Figure 6. As such they do require a minimum introduction for clarity and help interpretation of result.

In the initial revision of the manuscript other reviewers required clarification on the analytical pipeline of our WGCNA, and as such the authors feel that the level of detail in lines 274-277 is appropriate.

Discussion

Line 498 - "...conceivable that these cells increase in response to high parasitemia P. vivax to reduce the burden..." -> "...high P. vivax parasitemia..."

No clear limitation description

The authors could not understand this query.

Reviewer #3:

Authors have now addressed a very thorough set of referee comments. As a result, authors have greatly strengthened the scientific case for this manuscript. Not only the message is more clear and accurate now, but also the authors have produced additional and very valuable experimental data that support their conclusions. Therefore, I believe this piece of research is now ready for publication.

We thank this reviewer for their positive appraisal of the manuscript. Our study has been greatly improved by their input.

I would still suggest

1- Please, do make the source code for transcriptomic analysis available on GitHub.

The code and README are now on GitHub.

2- Move figure EV6 from supplementary to the main body of the manuscript.

Thanks for the suggestion. We contemplated that idea when submitting the revised version of the manuscript. However, the figure contains information relevant to both symptomatic and asymptomatic infection, we did not feel that it was a good addition to either Fig 4 or Fig 5. The authors feel that the figure mainly adds proof-of-concept material supporting the narrative of the manuscript and it is too small for a stand-alone main figure in a journal like MSB. We would like to follow editorial advice on what is the best fit for this figure.

22nd Jul 2025

Manuscript number: MSB-2024-12610RR

Title: Systems approach identifies monocyte imbalance in symptomatic and asymptomatic *P. vivax* malaria

Dear Prof Hansen,

Congratulations on an excellent manuscript, I am pleased to inform you that your manuscript has been accepted for publication in Molecular Systems Biology. Thank you for your comprehensive response to referee concerns. It has been a pleasure to work with you to get this to the acceptance stage.

Yours sincerely,

Poonam Bheda, PhD
Scientific Editor
Molecular Systems Biology
